# HIV-1 adapts to lost IP6 coordination through second-site mutations that restore conical capsid assembly

Alex Kleinpeter [1,3] ✉, Donna L. Mallery [2,3], Nadine Renner[2,3],
Anna Albecka [2], J. Ole Klarhof [2], Eric O. Freed [1] ✉ & Leo C. James [2] ✉

The HIV-1 capsid is composed of capsid (CA) protein hexamers and pentamers (capsomers) that contain a central pore hypothesised to regulate capsid assembly and facilitate nucleotide import early during post-infection. These pore functions are mediated by two positively charged rings created by CA Arg-18 (R18) and Lys-25 (K25). Here we describe the forced evolution of viruses containing mutations in R18 and K25. Whilst R18 mutants fail to replicate, K25A viruses acquire compensating mutations that restore nearly wild-type replication fitness. These compensating mutations, which rescue reverse transcription and infection without reintroducing lost pore charges, map to three adaptation hot-spots located within and between capsomers. The second-site suppressor mutations act by restoring the formation of pentamers lost upon K25 mutation, enabling closed conical capsid assembly both in vitro and inside virions. These results indicate that there is no intrinsic requirement for K25 in either nucleotide import or capsid assembly. We propose that whilst HIV-1 must maintain a precise hexamer:pentamer equilibrium for proper capsid assembly, compensatory mutations can tune this equilibrium to restore fitness lost by mutation of the central pore.

HIV-1 particle assembly is driven by the Gag precursor protein (Gag). Together with viral genomic RNA and a smaller number of GagPol precursor proteins, Gag assembles to form an immature lattice on the inner leaflet of the infected cell plasma membrane. Upon completion of assembly, the nascent virion buds off from the plasma membrane as an immature virus-like particle (VLP). This newly released VLP then undergoes maturation following cleavage of the Gag and GagPol precursors by the viral protease (PR), which is incorporated into virions as part of GagPol. A central feature of particle maturation is the condensation of the viral core, which contains the viral RNA genome and viral enzymes reverse transcriptase (RT) and integrase (IN) enclosed within a conical shell, or capsid, formed by the mature CA protein.

The conical HIV-1 capsid is built from quasi-equivalent hexamers and pentamers encoded by the CA protein. Each capsid contains approximately 250 hexamers and 12 pentamers, with 7 pentamers at the wide end and 5 at the narrow end, resulting in the capsid's characteristic cone shape[1]. The ability of CA to assemble into both hexamers and pentamers is essential for capsid assembly because, according to the principles of fullerene geometry, a completely enclosed, conical structure cannot be produced from hexamers only. Once thought to undergo rapid and spontaneous collapse after entry into a target cell, it is now clear that capsids remain largely intact and play a key role in all steps between fusion and integration[2,3]. The capsid is the principle interaction platform HIV-1 uses to interact with host co-factors early in infection[4-14]; it engages with microtubules to mediate transport to the nucleus[5] and, once delivered there, docks with the nuclear pore complex (NPC) via interactions between the capsid lattice and the phenylalanine-glycine (FG) repeats abundant in several NPC

[1]Virus-Cell Interaction Section, HIV Dynamics and Replication Program, Center for Cancer Research, National Cancer Institute, Frederick, MD 21702-1201, USA. [2]MRC Laboratory of Molecular Biology, Francis Crick Avenue, Cambridge CB2 0QH, UK. [3]These authors contributed equally: Alex Kleinpeter, Donna L. Mallery, Nadine Renner. ✉e-mail: alex.kleinpeter@nih.gov; efreed@mail.nih.gov; lcj@mrc-lmb.cam.ac.uk

proteins[14–19]. Interaction of the capsid with the NPC is necessary to ensure HIV-1 entry into the nucleus and integration into optimal regions of the host genome[20,21]. Remarkably, capsids are imported intact through the nuclear pore[22] and continue to interact with co-factors like CPSF6 inside the nucleus[23,24], until finally uncoating next to sites of integration[25]. The demonstration of intact capsid docking at the nuclear pore, together with its interaction with FG repeat pore proteins like Nup153, have also provided evidence for the hypothesis that the ability of HIV-1 to infect non-dividing cells is both a property of its capsid and the reason for the capsid's conical shape: A cone allows for directional transport through the nuclear pore, with its narrow end docking at the NPC[22] and increased FG interactions driving transport into the nucleus as the wider end of the capsid is drawn inward.

During its transport through the cell, HIV-1 reverse transcribes its RNA genome into double-stranded DNA, and the capsid is also important during this process. The capsid functions as a container for reverse transcription, maintaining both RT enzyme and RNA substrate at a high local concentration[26,27] and protecting the nascent viral DNA from innate immune sensing and degradation by host cell enzymes[28–31]. While encapsidation of the viral genome provides key advantages, an obvious disadvantage is that the viral transcriptional machinery is sequestered from the cytosolic dNTPs needed to fuel DNA synthesis. We previously reported that the hexamers and pentamers that comprise the capsid are porous, possessing a centrally located, positively charged channel through which dNTPs could pass[32]. This pore is lined with two rings of charged CA residues, R18 and K25 (Fig. 1A), and although permitting dNTP import, this clustering of positive charges is strongly destabilizing, raising the question of how capsids form in the first place and what keeps them from collapsing. It was subsequently shown that, in addition to dNTPs, the central pore of capsid hexamers and pentamers binds the polyanion inositol hexakisphosphate (IP6), which promotes capsid assembly and stability[33–35]. Such is the importance of IP6 in capsid assembly that HIV-1 uses its immature lattice to recruit IP6 during budding and packages it into particles[35–38]. IP6 availability may also be important in target cells to maintain stability, although depletion of IP6 in target cells does not significantly decrease wild-type (WT) infection[36,39,40]. This may be due to the ability of capsids to scavenge even low levels of the metabolite or to tolerate reduced IP6 occupancy, as HIV-1 mutants with unstable capsids show decreased infectivity in cells with reduced IP6 levels[41]. Structural evidence also supports the importance of IP6 in the assembly of properly formed capsids: The in vitro assembly of conical capsid-like particles (CLPs) is only possible in the presence of the polyanion[34], whilst viruses produced in cells depleted of IP6, or mutated to prevent IP6 packaging, fail to efficiently form mature capsids[37,38].

The severe infectivity defect resulting from mutation of either R18 or K25 supports the critical role of the charged rings formed by these residues in HIV replication[32,35,42,43]. The relative importance of each charged ring in IP6-mediated capsid stabilisation and dNTP binding and import is unknown, as mutating the rings in the mature capsid simultaneously destroys both IP6 and dNTP binding[32,35,42]. Moreover, it is unclear whether R18 and K25 are equally important or have non-redundant roles. R18 is essential for both nucleotide binding[32] and IP6-induced capsid assembly in vitro[34], whereas K25 is proposed to have more specialized roles in directing dNTPs into the capsid interior after capture by R18[43,44] or in allowing pentamers to form by stabilising closer packing through the coordination of a second IP6 molecule in addition to that bound by R18[45,46]. To address these key questions, here we have performed a series of forced evolution experiments to directly investigate the importance of R18 and K25 in HIV-1 replication. Our results show that HIV-1 can adapt to utilise a single ring of positively charged residues at position 18 but not a single ring at position 25. HIV-1 can compensate for the mutation of K25 through second-site mutations at either intra- or inter-capsomer interfaces, suggesting that K25 is not absolutely required for either capsid assembly or infection.

## Results

### Mutation of R18 and K25 profoundly reduces infection but differentially impacts capsid formation

To directly compare the importance of R18 and K25 in HIV-1 infection, we produced R18G and K25A mutant viruses and assessed their infectivity and capsid assembly in parallel. Based on structural models (Fig. 1A), R18G would be predicted to abolish binding of the upper IP6 molecule within the pore whereas K25A would be expected to abolish binding to the second, lower IP6 molecule. Both mutants displayed a profound loss of infectivity compared to WT virus (Fig. 1B). Consistent with this reduction in infectivity, assessment of capsid morphology in native virions by cryo-electron tomography (cryo-ET) revealed that, unlike WT, R18G and K25A possess either few or no regular capsid structures. Importantly however, the two mutants have strikingly contrasting assembly defects: No mature cores of any kind were observed for K25A (which we refer to as an under-assembly defect), with most particles containing unstructured density, whilst the majority of R18G virions contained capsid cores, albeit of an irregular rather than tubular or conical structure (Fig. 1C). Closer analysis also revealed that R18G virions often contained higher-order structures in addition to single capsids (an over-assembly defect) (Supplementary Fig. 1A). Both mutations also showed a modest decrease in viral production compared to wild-type (Supplementary Fig. 2). The fact that K25A has an under-assembly defect, while R18G suffers from over-assembly is consistent with the two residues playing different roles in capsid assembly and morphology. To explore these defects further, we compared the in vitro assembly kinetics of each mutant with WT capsid protein under either high salt or IP6-containing conditions. As previously described[47], R18G assembled into spheres in high salt (Supplementary Fig. 1B), whereas WT and K25A formed tubes[42]. Interestingly, although each mutant was strongly impaired for IP6-induced assembly at 100 μM CA and 1.25 mM IP6 (Supp Fig. 1C), at very high CA (900 μM) and IP6 (6 mM) concentrations, WT CA assembled predominantly into cones, whereas R18G assembled into spheres and a few tubes and cones, and K25A assembled exclusively into tubes (Fig. 1D). The spheres formed by R18G are similar in morphology and size (20 nm diameter) to the pentamer-rich spheres recently reported for capsid mutant M66A[46]. In contrast, tube assemblies are formed exclusively of hexamers[48]. A TVGG switch region in CA forms a $3_{10}$ helix in pentamers and a random coil in hexamers, thus one way R18G might favour pentamers is by increasing the propensity to form the $3_{10}$ helix[46]. However, as R18L has been shown to form pentamer-rich assemblies without refolding of the TVGG motif, it is also possible that R18G forms alternate pentamer structures[49]. Taken together, the data show that whilst both charged residues contribute to IP6-driven assembly, R18 is particularly important for hexamer formation and K25 for pentamers.

### K25A replication can be rescued by second-site mutations

Next, we asked whether R18 and K25 are absolutely required for viral replication or whether HIV-1 can evolve strategies to compensate for the defects induced by mutation of these residues. Propagation of R18 mutants (A,S,T,L,H,G,K) did not yield replication-competent virus in the highly permissive MT4 T-cell line (Fig. 2A & Supplementary Fig. 3). This supports the proposed importance of R18 in recruiting polyanions IP6 and dNTP that are needed for capsid assembly[34] and stability[35] and for encapsidated reverse transcription[32]. However, propagation of K25A in MT4 cells resulted in viral replication, albeit with delayed kinetics compared to WT (Fig. 2B). Peak replication for K25A was observed at ~day 20 post-transfection, compared to day 6 for WT, suggestive of the acquisition of second-site compensatory mutations (Fig. 2B). Re-passage of virus collected at peak replication in a variety of cell lines revealed that kinetics were least delayed in the most permissive cells, MT4, showed an intermediate delay in C8166 cells, and were most delayed in the least permissive cells, SupT1 (compare plots in Fig. 2C). Infected cells were collected at the peak of replication and

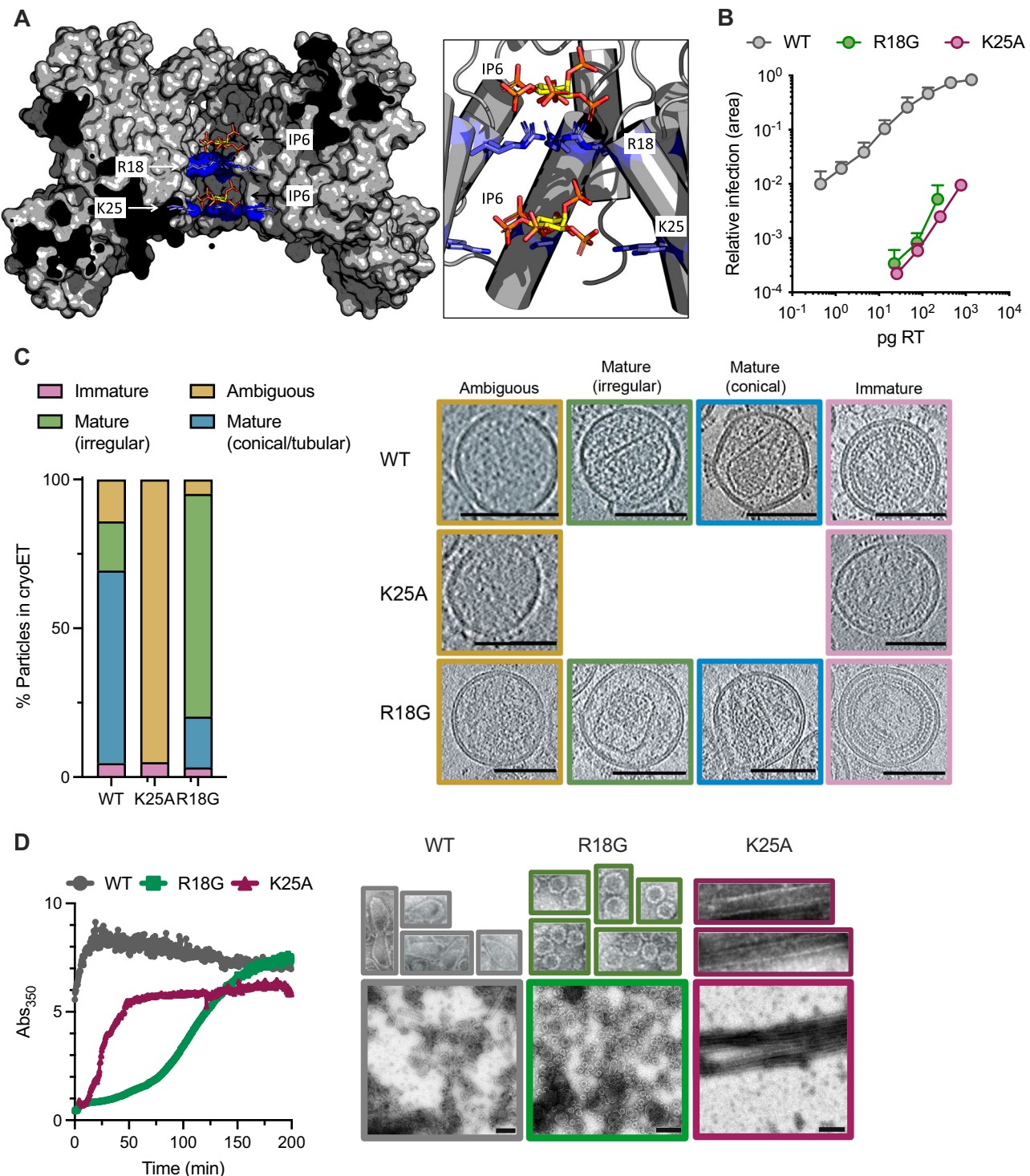

**Fig. 1 | Mutation of R18G and K25A profoundly reduces infection but differentially impacts capsid formation. A** Cross-section of a CA hexamer showing two IP6 molecules bound within the pore by the charged rings at R18 and K25 (based on 6R6Q). Side panel shows a close-up view. **B** Titration of WT and mutant HIV-1 VSV-G-pseudotyped virus on HEK293T cells with infectivity quantified as the proportion of infected cells (area of monolayer). Error bars depict mean ± s.e.m. from at least three independent experiments ($N = 3$). **C** Cryo-ET analysis of the indicated HIV-1 mutants.

Tilt series were collected and reconstructions performed to assess capsid morphology. A total of 51 WT, 38 K25A and 43 R18G particles were analyzed and classified into the indicated categories. Example sliced tomograms belonging to each category are shown together with the data for each virion. Scale bars, 100 nm. **D** In vitro assembly of 900 μM CA (WT, K25A and R18G) in 6 mM IP6, measuring absorbance of reaction over time at 350 nm. Negative stain EM images of assembly reactions are shown with 4x zoomed in sections displayed above. Scalebar: 200 nm.

genomic DNA was isolated. PCR-amplification of *gag* was performed, and amplicons were subjected to Sanger sequencing. Several distinct CA domain mutations were consistently observed (Supplementary Table 1), and these could be categorized into three main groups: Intercapsomer mutations such as G208R, T216I and G225S;

intracapsomer mutations such as N21S and A25T; and FG-binding site mutations A105T and T107I (Supplementary Table 1 & Fig. 3A). To determine whether the changes in CA are responsible for restoring viral replication, infectious molecular clones bearing these mutations in combination with either K25A or K25T were used to transfect MT4

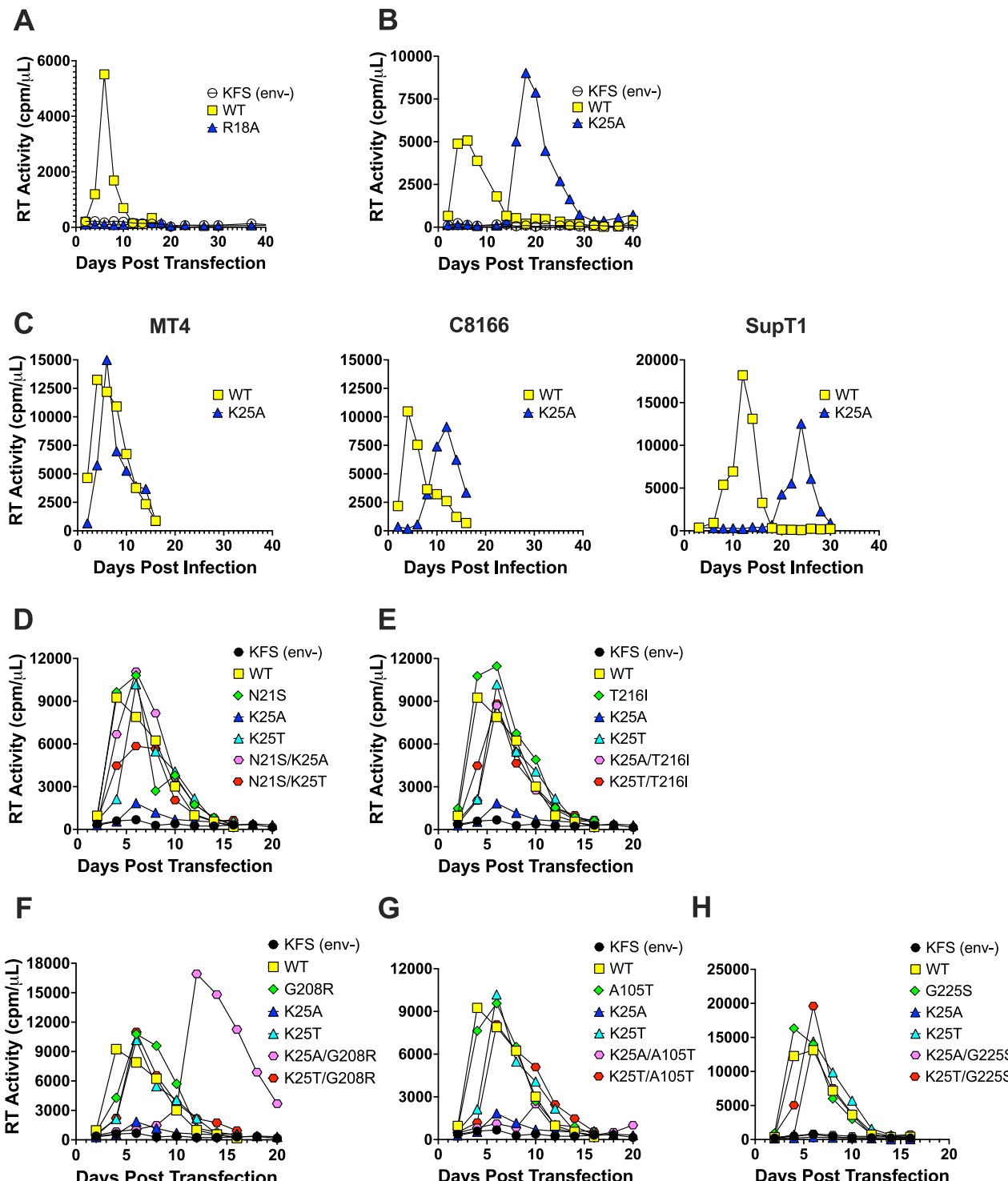

**Fig. 2 | K25A replication can be rescued by second-site mutations. A, B** MT4 cells were transfected with infectious molecular clones harboring mutations at either R18 (**A**) or K25 (**B**) and replication kinetics were assessed by quantifying supernatant RT activity. **C** Supernatants from (**B**) and similar repeat experiments were used to infect MT4, C8166, or SupT1 T cell lines to assess replication kinetics after acquisition of compensatory mutations. Infectious molecular clones of N21S (**D**), T216I (**E**), G208R (**F**), A105T (**G**) and G225S (**H**) variants were used to transfect MT4 cells to assess replication kinetics.

cells and their replication kinetics compared with those of WT. The addition of N21S and T216I restored WT replication kinetics to K25A (Fig. 2D, E), whereas G208R resulted in an intermediate delay and A105T and G225S had no effect (Fig. 2F–H). None of the second-site mutations appreciably altered K25T kinetics, which replicated similarly to WT.

To directly compare the ability of second-site mutations to restore virion infectivity, we produced a panel of mutant viruses and measured their single-cycle infection levels at a range of inputs (Supplementary Fig. 4A–D). All mutations increased virion infectivity when compared to K25A and some showed close-to-WT activity. Comparing infectivity across the entire panel of mutants at a single viral input

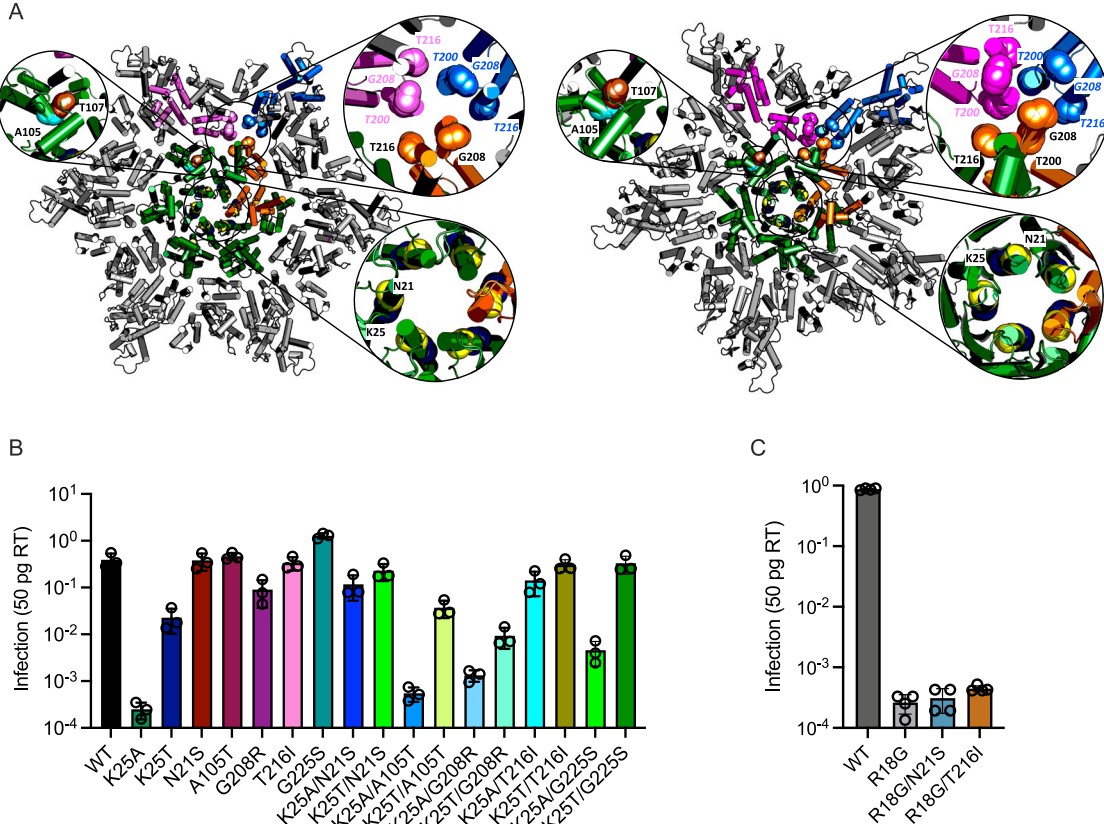

**Fig. 3 | Infectivity of K25A but not R18G can be rescued by second-site mutations. A** Representations of the hexamer (8CKV; left-hand side) and pentamer (8CKW; right-hand side) interactions in an HIV-1 capsid assembled in the presence of IP6[87]. The central hexamer or pentamer is shown in green and α-helices are indicated by cylinders. An example 3-fold inter-capsomer interface is highlighted with interacting monomers in orange, blue and pink. Zoomed-in regions show the location of second-site mutations selected during passage of K25A virus, with the specific residues labelled and indicated through sphere representation of their main-chain atoms. These regions correspond to the central pore (N21 and K25), the top of the CPSF6- binding pocket (A105 and T107) and the 3-fold interface (G208 and T216). **B**, **C** Single-round infectivity of the indicated viruses as measured by the proportion of infected cells (area of monolayer). Error bars depict mean ± s.e.m. from at least three independent experiments ($N = 3$).

revealed that some mutations such as N21S and T216I are capable of largely rescuing the defect in infection caused by K25A in isolation, whereas others, such as A105T, G208R and G225S only show a substantial rescue when combined with K25T (Fig. 3B). Notably, the ability of K25T to improve the infectivity of all second-site mutants, together with its much-improved infectivity as a stand-alone mutation when compared with K25A, suggests that the profound defect of K25A is caused in part by alanine being an unfavorable choice at this position rather than loss of the lysine per se. This contrasts with position 18 where a range of mutants were incapable of replication (Fig. 2A, Supplementary Fig 3). Nevertheless, given the ability of N21S and T216I to largely restore the infectivity of viruses lacking a charged residue at position 25, we tested whether they could rescue R18G. Neither mutation increased R18G infectivity (Fig. 3C), suggesting either that R18 is a more critical residue for HIV-1 infection than K25 or that R18 and K25 play separate roles in mediating capsid assembly and stability during virion maturation and infection.

## K25A compensating mutations restore DNA synthesis without altering nucleotide binding

None of the compensating mutations that arose during K25A propagation restores a positive charge at the centre of the pore. This is significant because K25 is thought to be required for binding pore ligands IP6 and dNTPs[42,43]. We therefore investigated whether IP6 and dNTP binding is affected by removal of K25 and/or addition of the intracapsomer compensating mutation N21S. Thermal stability measurements on crosslinked capsid hexamers using nanoDSF indicated

that the tested mutants were intrinsically less stable than WT, albeit to different degrees (Fig. 4A, B). Nevertheless, all mutants maintained the ability to bind IP5, IP6 and ATP and were stabilised to a similar extent as WT (Fig. 4A, B). Next, we investigated binding using a fluorescent ATP analog in a fluorescence polarization binding assay and found that both WT and mutant hexamers bound ATP with similar nanomolar affinity, with the exception of R18G which showed only micromolar binding (Fig. 4C). These results confirm that the primary binding site for pore ligands within hexamers is provided by R18 and suggests that the profound differences in infectivity observed between K25A, N21S/K25A and N21S/K25T are not the result of the loss and restoration of IP6 or nucleotide binding to hexamers.

Although the K25A mutation does not abolish nucleotide binding it is possible that without this second charged residue the directional movement of dNTPs through the pore and into the capsid is lost[43]. To test this, we compared reverse transcription during infection by viruses bearing the K25A mutation, alone or in combination with a compensating mutation located within or between capsomers (N21S or T216I, respectively). As observed previously[42], K25A was severely impaired for reverse transcription (Fig. 4D). In contrast, the viral DNA synthesis kinetics of K25A/N21S and K25A/T216I were similar to WT, with strong-stop transcripts peaking at ~4 h, first-strand transfer products at 6–8 h and second-strand transcripts at ~ 8 hrs (Fig. 4D). There was a difference in the overall levels of synthesized viral DNA, with K25A/N21S and K25A/T216I producing only ~20% of WT levels of second-strand transfer product, approximately proportional to the infectivity of these mutants ( ~ 20–30% of

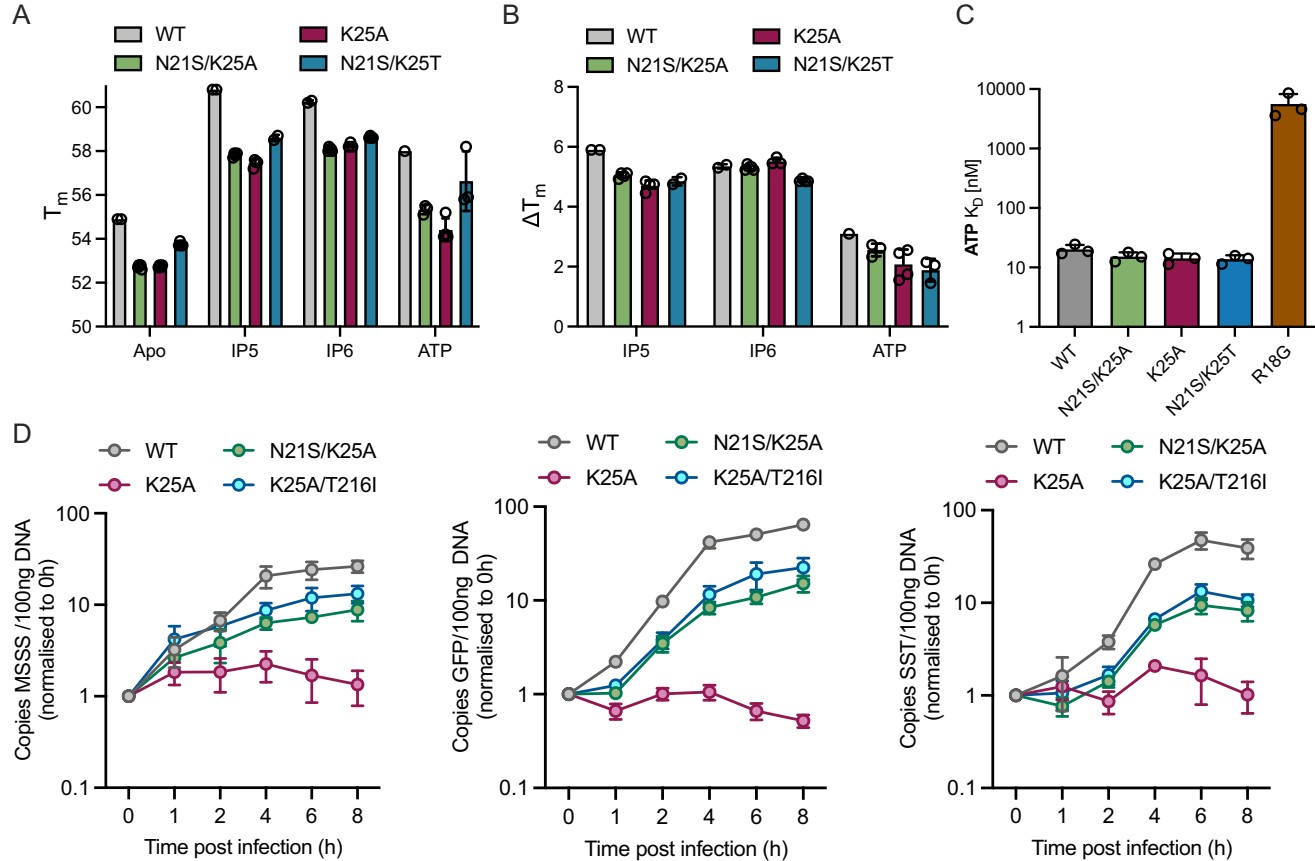

**Fig. 4 | K25A compensating mutations restore DNA synthesis without altering nucleotide binding. A** Thermostability of the indicated crosslinked CA hexamers either alone or in the presence of different polyanions, as measured by differential scanning fluorimetry. **B** Change in melt temperature ($\Delta T_m$) upon polyanion addition, calculated from the data in (**A**). Error bars depict the s.e.m. from three independent experiments. **C** Binding affinities of indicated WT and mutant crosslinked CA hexamers to fluorescent ATP as measured by fluorescence polarization binding assays. Error bars depict the s.e.m. from 3 independent experiments. **D** Quantification of viral DNA (vDNA) produced by the indicated viruses at various time-points post-infection (in hours). Early (minus-strand strong stop; MSSS), mid (GFP) and late (second-strand transfer; SST) products are quantified using specific primers. Error bars depict mean ± s.e.m. from at least three independent experiments ($N = 3$).

WT; Fig. 3B). These data suggest that K25 is not required for dNTP import through the pore.

## K25A in vitro capsid assembly is rescued by compensating mutations

By coordinating a second IP6 molecule in addition to that bound by R18, K25 is proposed to be important for capsid assembly; in particular, K25:IP6 binding has been suggested to be critical for the formation and stabilization of pentamers[45,46,50,51]. We therefore used an in vitro turbidity assay and negative-stain electron microscopy (EM) to investigate how K25A compensating mutations alter capsid assembly. First, we added 2.5 mM NaCl to WT, K25 A, K25A/T216I and N21S/K25A CA protein. Assembly occurred with similar kinetics for all mutants, except K25T and K25T/T216I, which assembled more rapidly than WT (Fig. 5A). Most reactions resulted in the formation of tubes, although N21S/K25A and N21S/K25T formed mainly spheres, while some cone-shaped capsids and spheres were observed for N21S and K25T/T216I (Fig. 5B). The shift to sphere assembly in proteins containing N21S suggests that this mutation favours pentamer formation. Notably, a previous report described sphere formation by an R18G/N21A mutant[52]. Next, we investigated whether compensatory mutations could rescue IP6-induced assembly of K25A cores. At IP6 and CA concentrations at which WT assembled quickly, only T216I rescued K25A assembly (Fig. 5C). In contrast, under the same conditions, all K25T-containing mutants were capable of assembling.

Repeating the assembly reactions with K25A and N21S/K25A at higher CA and IP6 concentrations resulted in assembly of the double mutant but not K25A (Fig. 5C). Negative-stain EM revealed that conical structures were present in most reactions, although K25T showed substantial tube formation and tubes were also found with the K25T-containing mutants N21S/K25T and K25T/T216I (Fig. 5D). N21S formed mostly spheres and occasionally small tubes (Fig. 5D). Within pentamers, N21 is located at a key inter-subunit interface formed by residues M39, V24 and K25 (Fig. 5E). Recent work has suggested that M39 acts like the pawl of a ratchet, switching between interactions with N57-T58 in hexamers and V24-K25 in pentamers[46]. Modelling N21S and K25T mutations at this interface suggests that they may favour the pentameric arrangement by reducing the steric constraints for ratchet movement and promoting the closer packing necessary in the pentamer (Fig. 5F, G). This is because mutations N21S and K25T result in smaller side chains in and around the ratchet, altering packing of 'pawl' residue M39. An allosteric network may connect the proposed ratchet with a TVGG motif between helices 3 & 4 and gate residue M66 (Fig. 5G), which together act as a hexamer/pentamer switch. The TVGG motif alters interactions at capsomer interfaces and sits behind the three-fold interface where second-site mutant T216I is located. It has been proposed that IP6 binding could regulate the TVGG switch though the exact mechanism is unclear. Residue K25 is thought to play a particularly key role in IP6 binding within pentamers, because binding of a second IP6

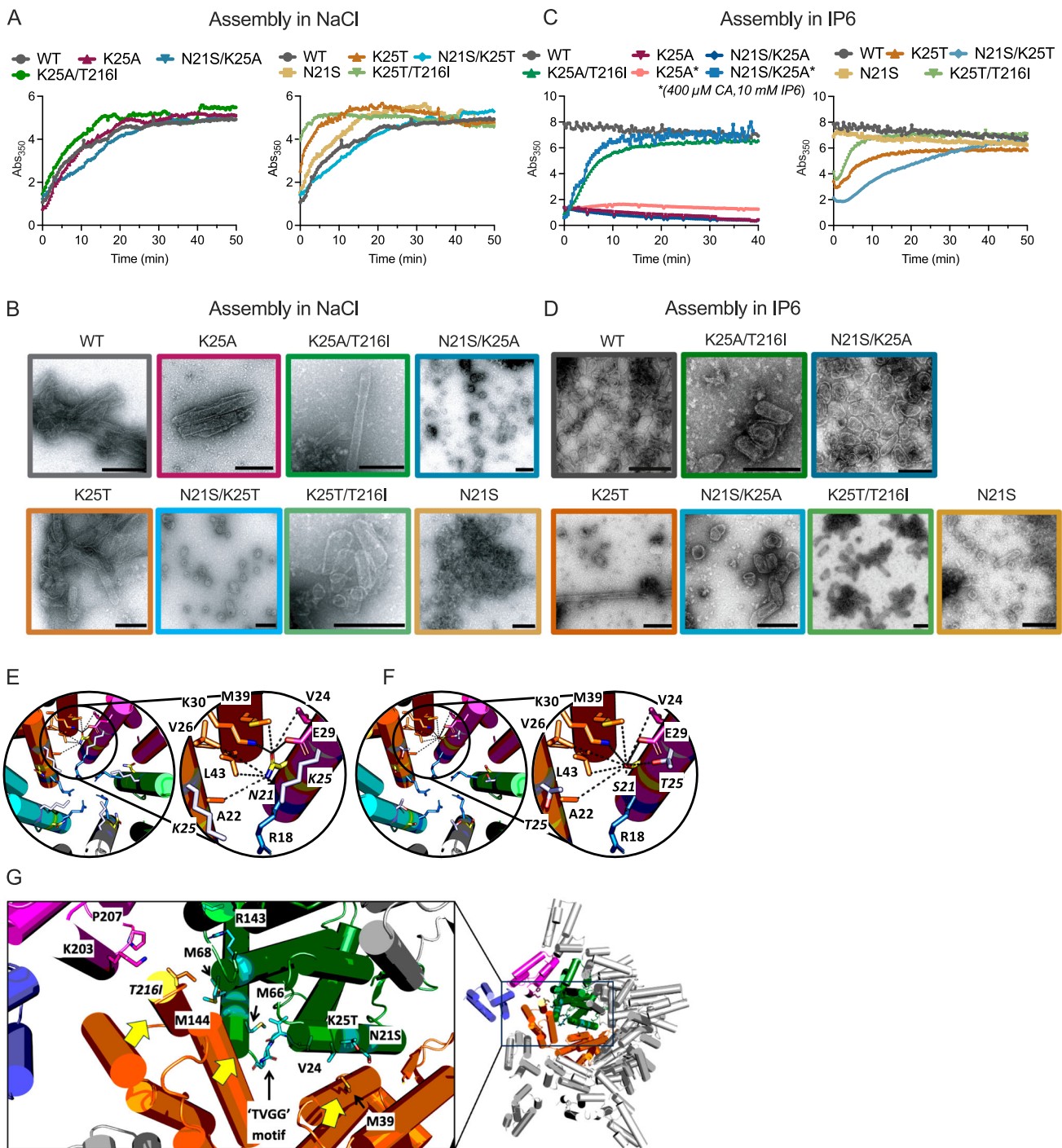

**Fig. 5 | K25A in vitro capsid assembly is rescued by compensating mutations.**
**A** Assembly of 200 μM K25A (left) or K25T (right) CA variants in 2.5 mM NaCl, measuring absorbance over time at 350 nm. **B** Negative stain EM images of assembly reactions from (**A**). **A**, **B** WT data are reproduced in each pair of graphs to allow easier comparison with mutants. **C** Assembly of 200 μM K25A (left) or K25T (right) variants in 1.25 mM IP6. Where indicated by the asterisk, assembly reactions were performed in 400 μM CA and 10 mM IP6. **D** Negative stain EM images of assembly reactions from (**C**). Scalebar: 200 nm. **E**–**G** Representations of packing interactions involving capsid pentamers (8CKW) in an HIV-1 capsid assembled in the presence of IP6[87]. **E** N21-proximal residues at the interface between neighbouring monomers in the pentamer, including the key pawl and ratchet residue

M39[46]. Dotted lines from N21 highlight the relative distance and positioning of nearby residues and do not necessarily indicate non-covalent interactions. **F** As (**E**), but with S21 modelled in place of N21 and T25 instead of K25. **G** Interaction networks that promote packing at multiple pentamer (8CKW) interfaces. Two monomers of the same pentamer are shown in orange and green and parts of monomers from adjacent capsomers in pink and blue. A proposed allosteric network connects a linear ratchet motif (V24, K25 & M49) to a 'TVGG' motif and gate residue M66 that modulate hexamer or pentamer formation[46]. The TVGG motif sits behind a three-fold interface where T216I is located. Yellow arrows indicate that interactions at each interface determine how closely monomers pack within and between pentamers.

molecule further down the pore allows closer packing of monomers. This may be the source of the propagated allosteric changes that favor a $3_{10}$ helix within the TVGG switch region[46]. Compensating mutations N21S and K25T could restore the closer packing that is lost with binding of the second IP6 molecule, whilst mutant T216I may alter the allosteric network connecting the various interfaces. Importantly, the fact that second site mutants can rescue pentamer formation without restoring binding of the second IP6 molecule suggests that monomer:pentamer:hexamer equilibrium is not solely determined by IP6.

### Second-site mutations stabilize K25A capsids that form inside virions

To test further the hypothesis that K25A-rescue mutations act by restoring pentamer formation, we used cryo-ET to examine capsid morphology in infectious N21S/K25A and K25A/T216I virions and compared this to our existing WT and K25A datasets. Particles were classified (or re-classified in the case of WT) into an expanded list of categories including immature, mature with conical, tubular or irregular capsids, or ambiguous. Remarkably, both intracapsomer mutant N21S and intercapsomer mutant T216I were sufficient to largely rescue mature capsid formation, displaying a distribution of particle types similar to WT (Fig. 6A, B). Because cryo-ET does not reveal whether assembled cores are stable without the protection of the viral envelope (eg post-fusion), we performed both TIRF microscopy and TRIM5α abrogation assays. For TIRF assays, we produced virus using a GagPol construct that has EGFP inserted between MA and CA and flanked by protease cleavage sites[53]. The EGFP is liberated during maturation of the virus and packaged non-specifically into the capsid, where it acts as a fluid-phase marker (Fig. 6C). EGFP-virions are immobilised and permeabilised with streptolysin O (SLO), as previously described[54,55]. By 30 min post-permeabilisation most WT capsids have undergone uncoating, as defined by a sufficient loss of capsid integrity to release the encapsidated EGFP signal (Fig. 6C–E). Addition of 50 μM IP6 inhibited uncoating, with over three-quarters of WT capsids remaining intact at the end of the experiment (Fig. 6D, E). In contrast, IP6 failed to prevent loss of EGFP from K25A, suggesting either that they do not form closed capsids or cannot be stabilised. Mutations N21S and T216I behaved similarly to WT and their capsids were stabilised by IP6. Importantly, addition of either mutation on top of K25A rescued the capsid-stabilising effect of IP6, with over two-thirds of capsids remaining intact after 30 min (Fig. 6D, E).

The above TIRF data was complemented by a TRIM5α abrogation assay. TRIM5α is a restriction factor that binds to the mature capsid and blocks infection[56]. In an abrogation assay, an increasing viral input is added to saturate the limited levels of TRIM5 inside the cell and restore infection. We used cells expressing rhesus macaque TRIM5α and co-infected them with a constant input of GFP-encoding reporter virus and different inputs of RFP-encoding mutant viruses. As the input of WT RFP virus was increased, there was an increase in GFP reporter virus infection, consistent with TRIM5α becoming saturated (Fig. 6F). In contrast, K25A RFP virus failed to saturate TRIM5α and minimal GFP reporter virus infection was observed. This is consistent with the cryo-ET data that K25A virions do not assemble mature capsids. However, both K25A/N21S and K25A/T216I RFP viruses were capable of saturating TRIM5α and driving increased GFP virus infection, confirming that the selected compensating mutations restore capsid formation and stability (Fig. 6F). Taken together, the in vitro assembly, cryo-ET, TIRF and TRIM5 abrogation data all suggest that N21S and T216I second-site mutations rescue K25A infectivity by restoring the ability to assemble a stable capsid. This highlights that maintaining K25 is not an intrinsic requirement for building an HIV-1 capsid capable of mediating infection.

## Discussion

The HIV-1 capsid contains hundreds of positively charged pores that bind dNTPs and IP6, providing a potential mechanism for the import of nucleotides for reverse transcription and driving capsid assembly and stability. Each capsid pore is lined with two rings of positively charged residues provided by R18 and K25 and each has been hypothesised to play a specific, non-redundant role[32,34,43–46]. In this study, we mutated R18 and K25 individually and performed forced evolution experiments to determine whether HIV-1 can overcome loss of either feature. We did not observe any replication of R18A/S/T/L/H/G/K virus or the emergence of viral variants containing rescuing mutations. This is consistent with the proposed role of R18 in building and stabilising capsids through the recruitment of IP6 and in importing dNTPs for encapsidated reverse transcription. Capsid-like particles can be assembled by mutants such as R18G (Fig. 1D) or R18L[49] but without the coordination of IP6 it is likely they are unstable[35]. In contrast to R18G and R18L, we observed the emergence of replicating K25A viruses that had acquired second-site compensating mutations and displayed nearly WT levels of infection. Thus, whilst a single ring of positively charged residues at position 18 can support HIV-1 replication, a single ring at position 25 cannot. This finding is consistent with sequence conservation amongst diverse retroviruses, as a charged residue at the structurally equivalent position to 18, but not position 25, is found in other lentiviruses such as FIV and in deltaretroviruses like HTLV-1.

The ability of HIV-1 to replicate without residue K25 is significant because K25 was thought to be essential for proper capsid assembly[45] and to direct nucleotide import during DNA synthesis[43,44]. Investigating how K25A infectivity is restored has revealed that a single second-site mutation either within capsomers (N21S) or at a capsomer:capsomer interface (T216I) is sufficient to rescue reverse transcription during infection. This suggests that K25 is not necessary for nucleotide import or DNA synthesis. However, the fact that the rescued K25A mutants still bind nucleotides and IP6 (Fig. 4A–C), use IP6 for in vitro capsid assembly (Fig. 5C, D), are stabilized by IP6 similar to wild-type (Fig. 6E) and have similar reverse transcription kinetics (Fig. 4D) suggest that the second-site compensatory mutations do not rescue K25A replication by conferring IP6 independence. Analysing the location of compensating mutations also does not suggest that they generate a new pore for nucleotide import. Compensating mutations that arose upon K25A propagation cluster into three main sites: within capsid pores (e.g. N21S, K25T), the FG-repeat binding interface (e.g. A105T, T107I) or at the three-fold axis between neighbouring capsomers (e.g. G208R, T216I and G225S). Interestingly, these positions are all hot-spots for viral adaptation and are commonly altered in response to selection pressures such as antiviral activity. More specifically, they are often the location of second-site suppressor mutations that don't confer resistance directly but restore fitness to resistant but less fit escape mutants. A105T is a suppressor mutation that appeared independently upon passage of cyclosporine-dependent mutants A92E[57] and T54A[58] and during selection for TRIM5 resistance[59]. T107I arose as a second-site mutation during selection for resistance to the maturation inhibitor GSK3532795[60] and the antiviral tripeptide GPG-NH$_2$[61], while T107N shows resistance to capsid inhibitor PF74[62]. A105T and T107I were also acquired as rescuing mutations during propagation of the PPIP motif mutant P122A[63]. G208R was selected as a compensatory mutation for NTD-CTD interface mutant H62F[64], during escape from both Mx2 inhibition[65] and capsid antivirals[66] and is resistant to SUN1 restriction[67]. Coincidentally, the same Mx2 study also observed the emergence of resistant mutant P207S, which is adjacent to T216I. T216I itself is a known second-site suppressor for unstable capsid mutant P38A[68] and rescues P38A sensitivity to IP6 depletion[41]. T216I confers resistance to PF74 and has been found in isolated transmitter founder viruses and 0.15% of HIV-1 group M strains in the Los Alamos HIV-1 database[69]. Finally, compensatory mutations at position G225 have

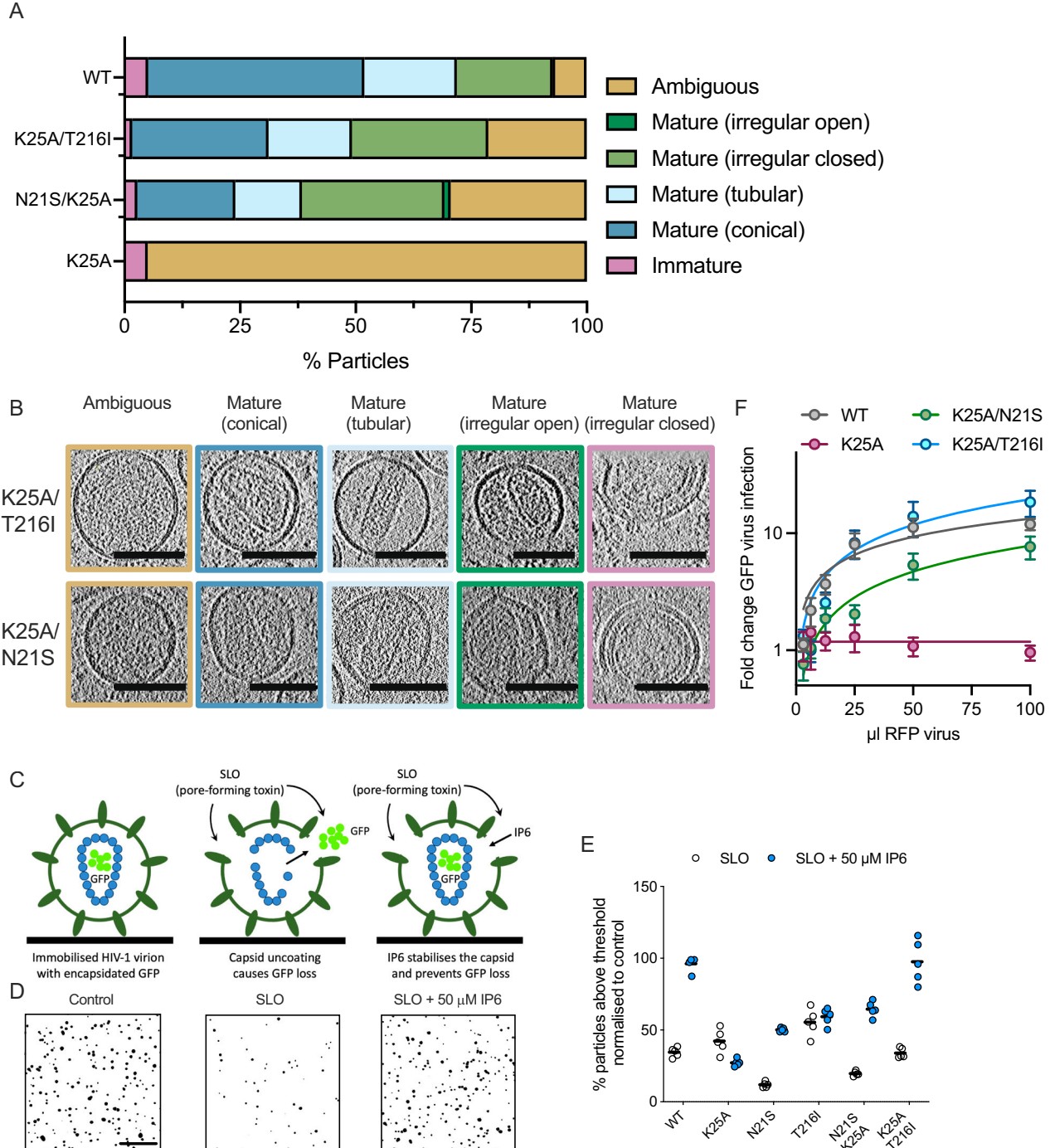

**Fig. 6 | Second-site mutations stabilize K25A capsids that form inside virions.**
**A**, **B** Cryo-ET of the indicated HIV-1 mutants. Tilt series were collected and reconstructions performed to assess capsid morphology. A total of 51 WT, 38 K25A, 146 N21S/K25A and 67 K25A/T216I particles were analyzed and classified into the indicated categories. Example sliced tomograms belonging to each category are shown together with the data for each virion. Examples of WT and K25A tomograms can be seen in Fig. 1. Scale bars, 100 nm. **C** Viruses containing EGFP as a capsid content marker were bound to glass dishes and permeabilised with SLO in the presence or absence of 50 μM IP6. Images were acquired with a TIRF microscope and particles counted using Fiji. **D** Representative masks generated during analysis of immobilised WT virus 30 min post-SLO or control

treatment ± addition of 50 μM IP6. Scale bar = 10 μm. **E** A single representative experiment (*N* = 1) where intact capsids from five 88 μm² images were counted, typically 500–1000/image for each condition, and the fraction of intact capsids in +SLO conditions plotted as a percentage of the mean under -SLO conditions (the overall mean of the fraction of intact capsids in +SLO conditions from five images is shown as a black bar ). **F** TRIM5 abrogation experiment. An increasing input of the indicated RFP reporter viruses was added to cells expressing rhesus TRIM5α in the presence of a constant dose of WT GFP virus. RFP viruses with a capsid capable of binding TRIM5 will saturate the restriction factor, leading to an increase in GFP virus infection. Error bars depict mean ± s.e.m. from at least four independent experiments (*N* = 4).

arisen during passage of maturation inhibitor (MI)-resistant and MI-dependent mutants. For instance G225S was selected during passage of the PF46396-dependent mutant P157S[70] and bevirimat-resistant SP1-A3V[71]. While MIs bind the immature Gag lattice and block CA-SP1

processing during assembly and release, their ability to block viral infectivity in the next round is attributable to the inability of virions released from MI-treated producer cells to assemble capsids during maturation. G225S restores infectivity to SP1-A3V without repairing its

CA-SP1 processing defect, suggesting that G225S may increase the efficiency of capsid assembly in the presence of lower levels of mature CA protein. G225S also arose during CTL-escape in chronic HIV-1 infection[72]. The common theme of these adaptation hot-spots is that they are sites of compensatory mutations that rescue infection by capsid stability. Restoring proper capsid stability is therefore likely the mechanism of action of the K25A-rescuing mutations reported here.

We also investigated how second-site compensatory mutations restore capsid assembly in the absence of K25. K25 recruits a second IP6 into capsid pores and this was proposed to be essential for pentamer formation, both in order to stabilise them[45] and induce allosteric changes at subunit interfaces to facilitate subunit packing[46]. Molecular dynamics simulations suggest a preference of IP6 for pentamers relative to hexamers and a faster rate of intercalation[50,51], while tomograms of native virions show stronger IP6 density for two molecules in pentamers than hexamers[73]; one coordinated by R18 and one by K25. The closer packing of these charged residues within pentamers likely means that pentamers are more unstable than hexamers in the absence of IP6 and more dependent upon binding two molecules. Nevertheless, comparing R18G and K25A mutants suggests that IP6 is important for both hexamers and pentamers because each mutation results in a different assembly phenotype: R18G forms pentamer-rich spherical cores whereas K25A forms hexameric tubes. The location of K25A second-site mutations, and their ability to shift K25A assembly from tubes to cones, suggests that they promote pentamer formation. Exactly how second site mutations promote pentamer formation is unclear but is likely through alterations at interfaces both within pentamers and across the lattice that make pentamers more energetically favourable. In the case of N21S, this may be through optimising contacts between monomers in and around the central pore. Closer packing is required in the pentamer between helices 1, 2 and 3 and the exchange of asparagine for serine may favour these interactions. N21S is also located next to the proposed ratchet that is thought to propagate changes allosterically to the 'TVGG' motif that switches between conformations favoring pentamer or hexamer NTD-NTD and NTD-CTD packing[46]. Mutant T216I may favour pentamers by promoting closer packing at the 2- and 3-fold symmetry axis in the capsid lattice. Although IP6 is proposed as a trigger for the 'TVGG' switch region, our data highlight that neither K25 nor the binding of two IP6 molecules is essential for pentamer formation or assembly of conical capsids. HIV-1 can restore K25A infectivity without reinstating the charged ring by subtly altering capsomer interfaces to stabilise pentamers and thus assemble a conical capsid.

Collectively, these data highlight an essential role for IP6 in assembling a stable HIV-1 capsid. The loss of K25:IP6 interaction prevents efficient pentamer formation, but also likely results in a broader decrease in the inherent stability of capsomers. We speculate that both of these defects contributes to the inability of K25A to form a stable capsid during maturation. The strongest K25A compensatory mutations, N21S and T216I, likely restore stable capsid assembly by reversing the K25A-induced defect in pentamer formation, whereas the suppressors A105T, T107I, G208R, and G225S play a more general (and milder) role in increasing capsid stability, as suggested by their appearance in multiple contexts (see above). The fact that mutations at multiple distinct locations across the capsid lattice can compensate for K25A argues that there are important allostertic networks linking the lattice interfaces together. Mapping these pathways and dissecting precisely the relative contribution of each interface (eg CACTD) to hexamer/pentamer equilibrium will require further structural work. Importantly, it should be noted that the compensatory mutations described here could also affect interactions with host factors that mediate or antagonize post-entry infection events. Mutations in the central pore (K25A/T and N21S), may alter interactions with FEZ1 and PQBP1, which are proposed to interact with the central pore to facilitate capsid trafficking to the nucleus and innate immune sensing

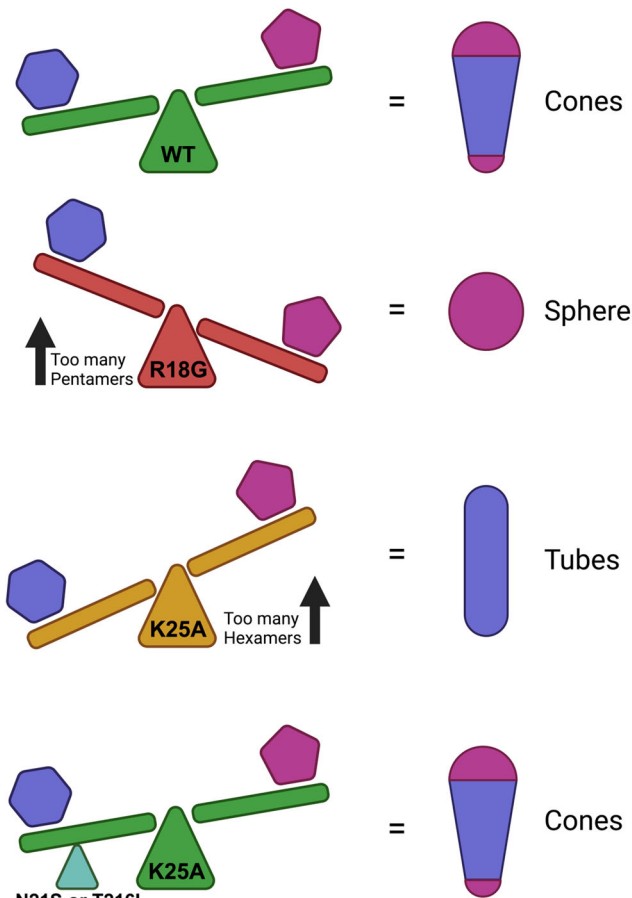

**Fig. 7 | Compensating mutations restore hexamer:pentamer equilibrium.** WT CA can form hexamers and pentamers but favours hexamers as more are needed to build a conical capsid. R18G shifts the equilibrium in favour of pentamers, resulting in the formation of pentamer-rich spheres in vitro. K25A shifts the equilibrium too far in favour of hexamers, resulting in the formation of hexameric tubes in vitro. Second-site compensatory mutations such as N21S or T216I stabilise pentamers, restoring WT equilibrium and thus conical capsid formation. Hexamers are blue and pentamers are purple. Note that in virions, R18G capsids are mostly irregular in their morphology whilst K25A capsids cannot be clearly assigned.

respectively[74–77]. Mutations near the FG-binding pocket (A105T and T107I) may influence interactions with Nup153 and CPSF6, each of which plays a crucial role in nuclear entry[78]. Finally, mutations near the 2- and 3-fold interface between capsomers may affect the ability of Mx2, which binds at this interface, to block nuclear entry[79–81].

We propose that HIV-1 maintains a specific hexamer:pentamer equilibrium that favours hexamers but allows pentamers to form (Fig. 7). Residues R18 and K25 play a key role in establishing this equilibrium through the coordination of IP6. Mutating R18 shifts the equilibrium in favour of pentamers by disrupting IP6 stabilization of hexamers, resulting in the assembly of pentamer-rich spheres. Conversely, mutating K25 shifts the equilibrium even further toward hexamers by disrupting IP6-mediated stabilization of pentamers, resulting in the assembly of hexameric tubes. Importantly however, HIV-1 has other ways to tune hexamer:pentamer equilibrium, as illustrated by the K25A compensatory mutations N21S and T216I described here, which restore WT capsid formation and thus infectivity (Fig. 7).

## Methods
### Cells and Plasmids
For replication and forced evolution experiments, pNL4-3, a lab-adapted, subtype B infectious molecular clone was used for

transfection of MT4 cells. The replication-incompetent pNL4-3 KFS (env-) molecular clone (Freed et al. PNAS. 1992) was used as a non-replicating control. MT4, C8166, and SupT1 human T cell lines were acquired from American Type Culture Collection (ATCC) and cultured in RPMI 1640 medium (Corning) supplemented with 10% fetal bovine serum (GenClone), 2 mM L-glutamine, penicillin (100U/ml; Gibco), and streptomycin (100µg/mL; Gibco). Replication deficient VSV-G pseudotyped HIV-1 virions were produced in HEK293T cells (CRL-3216, purchased from ATCC) using the packaging plasmid pMDG2, which encodes VSV-G envelope (Addgene plasmid # 12259), pNL4-3-derived pCRV GagPol (HIV-1 clade B)[82], and pCSGW[83] as described previously. HEK293T cells were cultured in Dulbecco's modified Eagle's medium (DMEM) with 10% FBS, 2mM L-glutamine, 100U/ml penicillin, and 100 mg/ml streptomycin (GIBCO) at 37 °C with 5% $CO_2$). Mutagenesis of CA was performed using the Quick-Change method (Stratagene) against pCRV GagPol. All cells were regularly tested to be mycoplasma free.

## Replication Kinetics

3µg of pNL4-3 DNA was mixed with 0.7 mg/ml DEAE dextran dissolved in Dulbecco's Phosphate Buffered Saline (DPBS) and this mixture was used to transfect $3 \times 10^6$ MT4 cells for 15 minutes at 37 °C. The transfection mix was removed by centrifugation and cells were resuspended in complete RPMI 1640 medium. Cultures were split (1:1) and supernatant samples were collected every two days for up to three months. To determine the kinetics of viral replication, HIV-1 reverse transcriptase (RT) present in supernatant samples was quantified using a previously described $^{32}$P-based assay[84].

## Forced evolution experiments

MT4 cells were transfected with NL4-3 derivatives harboring K25A amino acid substitutions (GCG or GCT Alanine codons) and supernatants were collected as described above. Cells and supernatant were collected at the peak of viral replication and genomic DNA was isolated using the QIAmp DNA Blood Mini kit (Qiagen - Catalog # 51106). Genomic DNA was used as template for PCR amplification of the HIV-1 Gag coding region. Gag fragments were treated with ExoSAP-IT PCR Product Cleanup Reagent (Thermo Fisher Scietific Catalog # 78201.1.ML) and submitted for Sanger sequencing. Sequencing results were evaluated using Snapgene. Viral supernatants from the peak of replication were quantified and $3 \times 10^5$ counts per minute (cpm) of virus were used to inoculate $3 \times 10^6$ fresh MT4, C8166, or SupT1 cells. Replication kinetics were measured to confirm reversion to wild-type replication and sequencing was repeated with these samples. Mutations identified were incorporated into pNL4-3 alone and in combination with K25A using a combination of site-directed mutagenesis (Q5 Site-Directed Mutagenesis Kit, *New England BioLabs*) and subcloning. Single- and double-mutant molecular clones were used to transfect MT4 cells to measure replication kinetics.

## Virus production & infection experiments

Viruses were produced from $2.5 \times 10^6$ cells in a 10 cm dish or $5 \times 10^5$ cells per well of a 6-well plate, plated the day before. Transfection mixtures were made using 200 µl OptiMem (GIBCO), 1 µg pMDG2, 1.5 µg pCSGW, 1 µg pCRV GagPol and 12 µl FuGENE6 (Promega). Mixtures were incubated at room temp for 15 min and then added in entirety to 10 cm dishes or 60 µl added to a well of a 6-well plate. Viral supernatants were harvested 48 hr post-transfection and filtered through a 0.45 µm filter and stored at −70 °C. For infection experiments with 293 T, cells were seeded at $0.75 \times 10^4$ cells per well into 96-well plates and left to adhere overnight. Indicated amounts of virus were added, and the plates were scanned every 8 h for up to 72 h in an IncuCyte (Satorius) to identify GFP-expressing cells.

## Virus Quantification

The level of RT enzyme was quantified using qRT-PCR as described previously with slight alterations[85]. In brief, 5 µl of viral supernatant was mixed with 5 µl lysis buffer (0.25% Triton X-100, 50 mM KCl, 100 mM Tris-HCl (pH 7.4), 40% glycerol) and 0.1 µl RNase Inhibitor and incubated for 10 min at room temperature before diluting to 100 µl with nuclease-free water. 2 µl of lysate was added to 5 µl TaqMan Fast Universal PCR Mix, 0.1 µl MS2 RNA, 0.05 µl RNase Inhibitor and 0.5 µl MS2 primer mix, to a final volume of 10 µl. The reaction was run on an ABI StepOnePlus Real Time PCR System (Life Technologies), with additional reverse transcription step (42 °C, 20 min).

## Virus particle production for tomography

Virus-like particles were produced in HEK293T as described above. Supernatants were harvested and passed through a 0.45 µm filter followed by a 0.22µm filter. The particles were concentrated by ultra-centrifugation over a 20 % (wt/vol) sucrose cushion (2 h at 28,000 rpm in a Beckman SW32 rotor; Beckman Coulter Life Sciences). The pellet was resuspended in PBS and incubated at 4 °C overnight to allow full resuspension.

## Cryo-Tomography

Virus-like particles were produced in HEK293T as described above. Supernatants were harvested and passed through a 0.45 µm filter followed by a 0.22-µm filter. The particles were concentrated by ultra-centrifugation over a 20% (wt/vol) sucrose cushion (2 h at 28,000 rpm in a Beckman SW32 rotor; Beckman Coulter Life Sciences). The pellet was resuspended in PBS. 10-nm-diameter colloidal gold beads were added to the purified HIV-1 mutants. 4 µl sample-gold suspension was applied to a glow discharged C-Flat 2/2 3 C (20 mA, 40 s). Grids were blotted and plunge-frozen in liquid ethane with a FEI Vitrobot Mark II at 15 °C and 100% humidity. Tomographic tilt series of WT, K25A, K25A/T216I, and K25A/N21S were acquired between −40° and +40°with increments of 3°, on a TF2 Tecnai F20 transmission electron microscope equipped with a Falcon III Direct Electron detector at 200 kV using Serial-EM under low-dose conditions at a magnification of 50000x and a defocus between −3 µm and −6 µm. Tomography of the R18G mutant was performed on a FEI Titan Krios transmission electron microscope at 300 kV equipped with a Gatan K2 summit direct electron detector and a Gatan Quantum energy-filter (GIF). Tilt series were acquired between −60° and +60° with increments of 3° using a dose symmetric scheme using Serial-EM[78]. Images were collected at a magnification of 33000x with 10 frames per tilt and a total dose of ~120 e−/Å2 across all of the tilts. Frames were aligned in SerialEM with a final pixel size of 3.667 Å per pixel in the unbinned image stacks. Tomograms were reconstructed using IMOD (4.9)[86]. The alignment of 2D projection images of the tilt series was performed using gold beads as fiducial markers, tomograms were reconstructed by back projection.

## Jess capillary protein detection system

Viral and cell samples were diluted to 5–20 pg/µl in PBS and separated by capillary electrophoresis using a separation module for 12–230 kDa (SM-W008) on Jess (Simple Western, Protein Simple) according to the manufacturer's instructions. Anti-HIV p24 (183-H12-5 C) was obtained from the NIH AIDS Reagent Programme, Division of AIDS, NIAID, NIH and was used at 1:100 dilution to ensure saturation and accurate quantification. Detection was via HRP using anti-mouse detection module (DM-002). Results were analysed using Compass software (Protein simple) and relative amounts of Gag cleavage products determined.

## TRIM5 Abrogation Assay

WT GFP virus and mCherry abrogating viruses were produced as described above. GFP virus was titrated onto FRhK cells at $5 \times 10^3$ cells/

well in 96 well plate in the presence of 5 mg/ml Polybrene, to determine the amount resulting in 1–3% infection. The quantity of mCherry viruses were measured by RT qPCR and equalised before titrating onto cells alongside a consistent amount of GFP virus (as above) with 5 mg/ml Polybrene. After infection GFP signal was measured by scanning every 8 h in an Incucyte (Sartorius) for 48–72 h and determining the cell area also positive for GFP.

## HIV RT in cells
293 T cells were plated into 24 well plates at $2.5 \times 105$ cell per well. GFP viruses were quantified and equalised, then added onto cells and plates spun at 380 g for 1 h at 4 °C. After 1 h media was removed and replaced with fresh media and samples harvested for time 0 h. Cells were then harvested, washed with PBS and frozen for downstream processing at the indicated time points. Control samples of uninfected cells and cells with boiled virus were harvested alongside the final time point. Total DNA was extracted from cells using the Qiagen DNeasy Blood and Tissue kit according to manufacturers instructions. Viral transcripts were quantified by qPCR of 2 µl extracted DNA with 5 µl TaqMan Fast Universal PCR Mix and 0.5 µl Primer:probe mix on an ABI StepOnePlus Real Time PCR System (Life Technologies). Early (minus-strand strong stop; MSSS), mid (GFP) and late (second-strand transfer; SST) products are quantified using specific primers: RU5 primers to detect strong-stop DNA (RU5 forward: 5′- GCCTCAATAAAGCTTGCCTTGA-3′; RU5 reverse: 5′- TGAC-TAAAAGGGTCTGAGGGATCT-3′; and RU5 probe 5′-(FAM) AGAGTCA-CACAACAGACGGGC (TAMRA)-3′), GFP primers to detect first-strand transfer products (GFP forward: 5′-CAACAGCCACAACGTCTATATCAT-3′; GFP reverse 5′-ATGTTGTGGCGGATCTTGAAG-3′and GFP probe 5′-(FAM) CCGACAAGCAGAAGAACGGCATCAA (TAMRA)-3′), and primers for second-strand transfer products (2ST forward: 5′- TGTG TGCCCGTCTGTTGTGT-3′; 2ST reverse: 5′- GAGTCCTGCGTCGAGA-GAGC-3′; and 2ST probe: 5′-(FAM)CAGTGGCGCCCGAACAGGGA (TAMRA)-3′).

## Protein production and purification
Capsid proteins were expressed in *E.coli* C41 cells for 4 h at 37 °C, lysed in lysis buffer (50 mM Tris-HCl (pH 8.0), 200 mM NaCl, 20% BugBuster, Protease inhibitor tablets, 1 mM DTT) and centrifuged (24,000 rpm, 1 h). The supernatant was precipitated with 25% ammonium-sulphate (wt/vol) followed by centrifugation (13,000 rpm, 20 min). The precipitated CA was resuspended and dialysed against 50 mM MES (pH 6.0), 20 mM NaCl, 1 mM DTT. The CA protein was further purified via a cation-exchange column with a gradient from 20 mM -1M NaCl followed by size exclusion chromatography with 50 mM Tris-HCl pH 8.0, 20 mM NaCl, 1 mM DTT, concentrated and snap frozen. The CA protein was expressed as described previously[38]

## Turbidity Assays
CA proteins were dialysed against 50 mM MES (pH 6.0), 40 mM NaCl, 1 mM DTT. CA proteins at a final concentration of 25–400 µM were mixed with NaCl (final concentration 2.5 M) or IP6 at 25 °C (final concentration 50 µM-2 mM). The apparent increase in absorbance reflecting increased light scattering ($OD_{350}$) was measured using a PHERAstar FSX Plate reader (BMG Labtech) in 384-well plate with shaking between each measurement at 25 °C or 37 °C.

## Negative stain
4 µl of sample from the assembly assay was put onto a glow discharged carbon coated grid (Cu, 300 mesh, Electron Microscopy Services), washed and stained with 2% Uranyl-acetate. Micrographs were taken at room temperature on a Tencai Spirit (FEI) operated at an accelerated voltage of 120 keV and Gatan 2k × 2 k CCD camera. Images were collected with a total dose of ~30 e⁻/Å² and a defocus of 1–3 µm.

## Nanoscale differential scanning fluorimetry (NanoDSF)
DSF measurements were performed using a Prometheus NT.48 (NanoTemper Technologies) over a temperature range of 20–95 °C using a ramp rate of 2.5 °C/min. CA hexamers were mixed with equimolar concentrations of IP5, IP6 or ATP to a final concentration of 100 µM.

## Particle stability by TIRF
pCRV GagPol iEGFP was generated by inserting EGFP between the MA and CA of Gag, flanked by protease cleavage sites so that EGFP is released from the Gag upon maturation[53]. To produce iEGFP containing viruses, cells were transfected as described above with 1 in 4 of pCRV GagPol being replaced with pCRV GagPol iEGFP. For generation of capsid mutants both pCRV GagPol and iEGFP version of the plasmid contained relevant capsid mutations. 6 hours after transfection cells were washed and then incubated for 3 days with 2% FBS containing media. Supernatants were then filtered, diluted 1 in 5 in SLO reaction buffer (SRB, 100 mM Tris-HCl pH 8, 150 mM NaCl) and then placed on 8-well glass Ibidi dishes precoated with poly-L-lysine (Sigma, P4707) for binding. After one wash viruses were incubated for 30 min with 20 nM SLO in the presence or absence of 50 µM IP6 in the SRB. Untreated wells were used as control for normalisation. After incubation wells were washed 2 times with SRB +/− IP6 and then fixed with 4% formaldehyde for 20 min. Images were acquired with a Nikon TIRF inverted microscope with a 100x/1.49NA oil-immersion objective, a 1.5x intermediate magnification and Prime95B sCMOS camera from Photometrics resulting in a 74 nm pixel size. Particle numbers were analysed in Fiji, where images were median filtered, and background subtracted. An intensity threshold was used to create a mask and a watershed step allowed separation of touching particles. ROI were filtered by area within 10 to 150 pixels.

## Streptolysin O (SLO) production
54 kDa (residues 101−571) version of the SLO was tagged with 6xHis by cloning into pOPTH-Tev vector for production in *E.coli* C41 strain. Cells were lysed with 20% BugBuster in 25 mM Tris pH8, 150 mm NaCl, 1 mM DTT, benzonase and lysozyme. Lysate was loaded onto a Ni column in 25 mM Tris pH8, 150 mM NaCl, 1 mM DTT, 10 mM Imidazole, washed with buffer with 10 mM Imidazole, followed by 40 mM Imidazole and then eluted with 200 mM Imidazole. 1 ml fractions were then checked on a polyacrylamide gel, peak fractions combined and dialysed to 25 mM Tris pH8, 150 mM NaCl and 1 mM DTT. Protein was then aliquoted and stored at −70 °C.

## Statistical analysis
Unless otherwise indicated, statistical analyses were Student's t-tests and performed using GraphPad Prism 9 software (GraphPad). Error bars depict the mean +/− SEM unless indicated otherwise.

## Statistics and reproducibility
All experiments were repeated independently at least three times, unless otherwise specifically stated. The exact number of repeats (N), determination of error and statistical analysis is given for each specific experiment in the figure legends and the primary data is provided in Source Data.

## Reporting summary
Further information on research design is available in the Nature Portfolio Reporting Summary linked to this article.

# Data availability
All data are available in the accompanying Source Data files. Source data are provided with this paper.

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

## Acknowledgements

This work was supported by the MRC (UK; U105181010), a Wellcome Trust Investigator Award (200594/Z/16/Z), a Wellcome Trust Collaborator Award (214344/A/18/Z) to LCJ. Research in the Freed laboratory is supported by the Intramural Research Program of the Center for Cancer Research, National Cancer Institute, National Institutes of Health. AK was supported in part by an Intramural AIDS Research Fellowship and the National Institute of Allergy and Infectious Diseases (K99AI174891-01). We would like to acknowledge our collaborative interactions with the Pittsburgh Center for HIV Protein Interactions (U54AI170791) and the Behavior of HIV in Viral Environments Center (U54AI170855). We are grateful to the MRC-LMB Electron Microscopy Facility for access and support of electron microscopy sample preparation and data collection.

## Author contributions

Study was conceived by AK, DLM, EOF & LCJ. Manuscript was written by LCJ with contributions from AK & EOF. Experiments were performed by AK, NR, DLM, AA, JOK. Analysis was carried out by all authors.

## Competing interests

The authors declare no competing interests.
