## [Peer Review File · Nature Communications]

HIV-1 adapts to lost IP6 coordination through second-site mutations that restore conical capsid assemblyREVIEWER COMMENTS

Reviewer #1 (Remarks to the Author):

In their article “HIV-1 adapts to lost IP6 coordination through second-site mutations that restore conical capsid assembly,” Kleinpeter et al. performed a series of forced evolution experiments to investigate the specific roles of R18 and K25 pores in the CA assembly. To compare the importance of R18 and K25 rings, the authors compared the R18G and K25A mutant infectivity and assembly behavior. K25A but not R18 mutations can be restored by second-site mutations, indicating that K25 is pivotal yet substitutable, but R18 is indispensable for HIV-1 infection. The authors also provide evidence that neither K25 nor the binding of two IP6 molecules is essential for assembly of pentamers. The manuscript presents several important observations relevant to HIV-1 biology. Before recommending for publication, some points should be clarified.

Major comments:

- Page 7- “Propagation of R18 mutants R18A, R18S or R18T did not yield replication-competent virus in the highly permissive MT4 T-cell line...”

Can the authors comment on why R18A, R18S or R18T mutants did not yield replication-competent viruses? Maybe these mutants do not produce mature-like capsids or fail to enclose viral RNA. In a recent study by Schirra, R. T., dos Santos, N. F., Ganser-Pornillos, B. K., & Pornillos, O. *bioRxiv* (2024), Arg18 substitutions reveal the capacity of the HIV-1 capsid protein for non-fullerene assembly. The R18L mutant has been shown to produce WT capsid like particles. The authors could investigate whether propagation of R18L mutant can produce replication-competent virus in the MT4 cell line.

- Page 6- “whereas R18G assembled into spheres and a few tubes and cone...”

The TVGG molecular switch apparently modulates the ability of CA to form pentamer or hexamer. The R18L mutant has been shown to form pentamer without refolding of the TVGG switch. The authors could comment on likely TVGG conformations in the spherical particles under both at high salt and IP6 conditions for R18G.

- Page 7- “to Sanger sequencing. Several distinct CA domain mutations were consistently observed... Intercapsomer mutations such as G208R, T216I and G225S...”

The authors have mentioned that mutation at G225 was observed during K25A propagation. Yet there is no information on the replication kinetics and infectivity of the G225S mutant. The authors could comment on why G225 mutants were not considered for replication kinetics and infectivity studies.

- Fig 2D- Comment on why N21S and K25T mutations restore WT-level RT activity yet N21S/K25T mutant RT activity is less than N21S/K25A?

The authors have mentioned that The T216I mutation may influence M66 side-chain dynamics as it may form a hydrophobic pocket with M68 and M144 located directly behind the gate residue (Figure 5G) – This is an important statement and currently only briefly stated in the manuscript. The authors could expand on the relationship between IP6 binding and pentamer/hexamer thermodynamic equilibrium.

Modelling N21S and K25T mutations at this interface suggests that they may favor the pentameric arrangement by reducing the steric constraints for ratchet movement and promoting the closer packing necessary in the pentamer (Figure 5F). – This is also an important statement, and the

authors could provide additional mechanistic insight (e.g., additional text and supporting figure) on what is the network of interactions that promotes closer packing in the pentamer.

Minor comments:

- Fig 2- legend “K25A but not R18 infectivity can be rescued.”

Do the authors mean R18 infectivity or R18 mutant (R18G) infectivity?

- Page 6- "K25A but not R18A/S/T infectivity can be rescued by second site mutations."

Considering the differential role of R18 and K25 in capsid assembly, it is not surprising that the second site mutations responsible for rescuing K25A could not restore R18 mutant infectivity. Maybe the authors can clarify if secondary mutations other than N21S/R18G or T216I/R18G can restore the effects of R18 mutations?

Reviewer #2 (Remarks to the Author):

HIV-1 capsid is an assembly of 250 hexamers and 12 pentamers. Hexameric (also pentameric) capsomeres assemble a centralized pore that is bound by IP6 and has been postulated to act as a nucleotide importer by the James group. The formation of the central pore requires the coordination of 2 amino acids R18 and K25. Both amino acids coordinate the binding of capsid to IP6, which has been shown to stabilize the overall assembled capsid. The integrity of the capsid which is supported by IP6 binding is a key aspect that is necessary for post-entry roles of capsid, which includes serving the role of a container where protected viral DNA synthesis by reverse transcription could take place.

Here, Kleinpeter. et al present an elegant study that uses the forced evolution of viruses to compensate for the IP6 binding site mutant at the R18 and K25 amino acid residue of CA, the protein that makes up the HIV-1 conical capsid shell. Previously postulated importance of these residues is tested here elegantly, by forcibly evolving viruses under selective pressure in a replication assay. The authors conduct a series of structural, biochemical, and virological studies to illustrate the role of the second site suppressors on capsid assembly (in vitro, and in virions), capsid stability after membrane stripping and in cells. While R18 mutants failed to replicate, the K25A mutants evolved more efficiently to replicate in cultured cells by acquiring second-site suppressor mutants in the intra- and inter-capsomere interfaces. These compensatory substitutions in the K25 mutants overcome infectivity defect, while substitutions with T is more favorable than A at position K25. They find that R18A/T/S is a more rigid mutant, that fails to recover in replication assays. The reason why R18 mutants fails to replicate is likely because this mutation leads to predominantly pentameric assemblies, which is nicely demonstrated and supported by complementary structural studies conducted in this work. On the other hand, the acquisition of compensatory second-site suppressor mutations in K25A mutant viruses facilitate the assembly of conical capsids.

The main conclusion that R25 which was originally thought to be strictly essential can be by passed by second site suppressors on CA is well supported by the data and appropriately described in the main text. The broader implications includes 'conical morphology', and thus an optimal hexamer:pentamer equilibrium is critical for the assembled HIV-1 capsid functions in mediating virus infection.

Notable questions that arose from my review include:

(1) How does IP6 fit into this current story? Is IP6 even required for HIV-1 infection? Given that K25A/T216I among other mutants can effectively infect cells, do they still require IP6 binding and does dNTP import into the core of the virus still go through the central pore formed in K25A/T216I mutant hexamer? Do the authors perceive that dNTP import into the core can also occur through other mechanisms? It will be nice to see endogenous reverse transcription (ERT) in the identified new capsids.

(2) R18G forms pentameric capsids, and K25A forms hexameric capsids – so R18 is required for hexamer formation and K25 for pentamer formation. The failure of R18 mutants to replicate highlights importance of hexamers in capsid assembly. However, how does secondary suppressor mutations in K25A/T mutants restore pentamer formation? The answer to this question, while is somewhat addressed by using models to show proximity of second-site suppressors to the K25 amino acid, this is at best hypothetical.

(3) Additional question that arose: Does the central hexamer/pentamer ring coordinating IP6 still formed following K25A/T + T216I substitutions? And is IP6 still required to stabilize these capsids? remains unclear. I say this because in Fig. 6E, while IP6's role in stabilization of WT, N21S, N21S/K25A is somewhat shown (although distribution seems weird, in all of these the fold-difference of IP6 stabilization appears similar), there seem to be less of an effect of IP6 on T216I, and K25A/T216I. I would encourage the authors to consider re-plotting this graph as % of intact cores at 30 minutes by normalizing to initial intact cores seen immediately after SLO treatment (% of initial timepoint), and plotting bar graphs of fold-differences of IP6 stimulated stability changes may provide more insights and better interpretations. Also, I am not sure the TRIM5 abrogation assay reports stability of capsids in cells, but rather TRIM binding capacity.

(4) Regarding completeness of the implications of new substitutions: While infectivity measurements are provided to show that second-site suppressors can overcome K25A mutation defects, what happens to the second-site suppressor containing K25A capsids in cells remain unclear. The cell biology aspect of these new capsid variants is missing. The kinetics of viral DNA synthesis is shown seems like no apparent change, how about nuclear import, and integration? Are there effects in primary cells in terms of sensing where host-factor usage is critical? It is known that R18A cannot bind a host-factor PQBP1, does K25A/T216I or in combination with N21S impose defects in this interactions?

Minor:

Pg.3 Line 43 "200 hexamers" is this a correct number? Perhaps provide a citation. Published literature suggests 240 or 250 hexamers.

Pg. 3 Line 48: "capsids remain intact" I would suggest using the phrase "remain largely intact" here, since this is still an unresolved issue.

Pg. 6 Line 120-123: "The fact that K25A has an under-assembly defect, while R18G suffers from over-assembly is consistent with the two residues playing different roles in capsid assembly and morphology" – is it possible these mutants affect virus budding and release? Please include data for virus production and associated western blots

Pg. 8 Line 185: “fluorescent ATP analog” please describe here how this incorporates into the thermal stability measurements. Why fluorescent? Can ATP alone be used?

Pg. 8 Line 201-203: “These data suggest that K25 is not required for dNTP import through the pore, although the inability of second-site mutations N21S or T216I to fully restore the accumulation of viral DNA to WT levels suggests that either import is slightly less efficient, or capsids are slightly less stable.” Check for sense. To this reviewer the suppressor mutants seems to recover defects in vDNA synthesis – fully and is correlated to infectivity data. Not sure what is being highlighted here.

Line 315: T216I was also isolated from transmitter founder viruses and in patient samples (Siddiqui et al.,), is it not interesting that this mutant also shows up in the forced evolution?

Fig. 6C-E: Core stability assay using iGFP: “Capsid particles from three to five images were acquired for each condition and normalised to untreated control” please elaborate. (1) was the particle distribution similar when imaging intact virions, (2) how many iGFP containing cores were retained following SLO treatment, and (3) what was the % that remained stable from this pool.

From Fig. 6C-E legends: “Graph shows particle numbers from multiple images from three independent experiments. Multiple unpaired T tests were used to compare conditions to IP6.” - As of now, I see a distribution plot of particle numbers, but what seems like a distribution plot of fluorescence. If the numbers were used from independent experiments one would expect to see a few dots and error bars.

The in vitro measurements of stability seem to suggest that K25A is just as stable as the WT capsid (Fig. 6E). Yet it fails to overwhelm TRIM restriction in cells (Fig. 6F). Does K25A fail to abrogate TRIM activity because it does not bind the restriction factor? I am unsure if this is the right assay to demonstrate capsid stability in cells. Also, a control using unstable capsids seems to be missing.

Reviewer 1

In their article “HIV-1 adapts to lost IP6 coordination through second-site mutations that restore conical capsid assembly,” Kleinpeter et al. performed a series of forced evolution experiments to investigate the specific roles of R18 and K25 pores in the CA assembly. To compare the importance of R18 and K25 rings, the authors compared the R18G and K25A mutant infectivity and assembly behavior. K25A but not R18 mutations can be restored by second-site mutations, indicating that K25 is pivotal yet substitutable, but R18 is indispensable for HIV-1 infection. The authors also provide evidence that neither K25 nor the binding of two IP6 molecules is essential for assembly of pentamers.

The manuscript presents several important observations relevant to HIV-1 biology. Before recommending for publication, some points should be clarified.

Major comments:

Q1. Page 7- “Propagation of R18 mutants R18A, R18S or R18T did not yield replication competent virus in the highly permissive MT4 T-cell line...” Can the authors comment on why R18A, R18S or R18T mutants did not yield replication competent viruses? Maybe these mutants do not produce mature-like capsids or fail to enclose viral RNA. In a recent study by Schirra, R. T., dos Santos, N. F., Ganser-Pornillos, B. K., & Pornillos, O. bioRxiv (2024), Arg18 substitutions reveal the capacity of the HIV-1 capsid protein for non-fullerene assembly. The R18L mutant has been shown to produce WT capsid like particles. The authors could investigate whether propagation of R18L mutant can produce replication-competent virus in the MT4 cell line.

A1. Our explanation for the lack of R18A, R18S or R18T replication is that the positive charge provided by the arginine is crucial and cannot be rescued by a non-charged residue at this position. This is consistent with work from multiple labs that an arginine is needed to bind IP6 and dNTPs. These polyanions are proposed to promote assembly and stability of the capsid and allow encapsidated reverse transcription (see Dick et al. 2018 Nature, Mallery et al. 2018 eLife and Jacques et al. 2016 Nature). It is possible to form capsids or capsid-like particles with R18 mutants, as shown by us in Figure 1 and in the referenced bioRxiv study but these would be predicted to be unstable and unable to carry out efficient DNA synthesis.

We have followed the reviewers suggestion and introduced the following comments into the Results: “This supports the proposed importance of R18 in recruiting polyanions IP6 and dNTP that are needed for capsid assembly[34] and stability[35] and for encapsidated reverse transcription[32].”

And the Discussion: “This is consistent with the proposed role of R18 in building and stabilising capsids through the recruitment of IP6 and in importing dNTPs for encapsidated reverse transcription. Capsid-like particles can be assembled by mutants such as R18G (Figure 1D) or R18L[49] but without the coordination of IP6 it is likely they are unstable[35].”

We have also propagated additional R18 mutants (L,H,G,K) in long-term replication assays and did not observe any replication. This data has been included as new Supplementary Figure 3.

Q2. Page 6- “whereas R18G assembled into spheres and a few tubes and cone...” The TVGG molecular switch apparently modulates the ability of CA to form pentamer or hexamer. The R18L mutant has been shown to form pentamer without refolding of the TVGG switch. The authors could comment on likely TVGG conformations in the spherical particles under both at high salt and IP6 conditions for R18G.

A2. This is an interesting point and we thank the reviewer for raising it. We have introduced the following into the Results section: “A TVGG switch region in CA forms a 3_{10} helix in pentamers and a

random coil in hexamers, thus one way R18G might favour pentamers is by increasing the propensity of CA to form the 3_{10} helix[46]. However, as R18L has been shown to form pentamer-rich assemblies without refolding of the TVGG motif, it is also possible that R18G forms alternate pentamer structures[49].”

Q3. Page 7- “to Sanger sequencing. Several distinct CA domain mutations were consistently observed... Intercapsomer mutations such as G208R, T216I and G225S...” The authors have mentioned that mutation at G225 was observed during K25A propagation. Yet there is no information on the replication kinetics and infectivity of the G225S mutant. The authors could comment on why G225 mutants were not considered for replication kinetics and infectivity studies.

A3. We have now included additional data on G225S for completeness (Figure 2H & 3B). Unlike N21S and T216I, G225S does not restore K25A replication kinetics and only modestly increases infectivity in a single-round.

Q4.1. Fig 2D- Comment on why N21S and K25T mutations restore WT-level RT activity yet N21S/K25T mutant RT activity is less than N21S/K25A?

A4.1. We thank the reviewer for pointing out this difference in the replication kinetics curves for N21S, K25T, and N21S/K25T. However, we generally do not consider the height of the replication curves when comparing viral replication kinetics. We find that the RT activity at the peak of replication can vary significantly between experiments. The most important takeaway is where the peak of replication occurs relative to other virus samples in the same experiment. In this case, the replication of each of N21S, K25T, and N21S/K25T peaks at ~Day 6.

Q4.2 The authors have mentioned that The T216I mutation may influence M66 side-chain dynamics as it may form a hydrophobic pocket with M68 and M144 located directly behind the gate residue (Figure 5G) – This is an important statement and currently only briefly stated in the manuscript. The authors could expand on the relationship between IP6 binding and pentamer/hexamer thermodynamic equilibrium. Modelling N21S and K25T mutations at this interface suggests that they may favor the pentameric arrangement by reducing the steric constraints for ratchet movement and promoting the closer packing necessary in the pentamer (Figure 5F). – This is also an important statement, and the authors could provide additional mechanistic insight (e.g., additional text and supporting figure) on what is the network of interactions that promotes closer packing in the pentamer.

A4.3 We have followed the reviewer’s suggestion and significantly expanded this section of the manuscript. We have generated a new supporting figure that replaces 5G and added the following text to the Results section:

“Modelling N21S and K25T mutations at this interface suggests that they may favour the pentameric arrangement by reducing the steric constraints for ratchet movement and promoting the closer packing necessary in the pentamer (Figure 5F&G). This is because mutations N21S and K25T result in smaller side chains in and around the ratchet, altering packing of ‘pawl’ residue M39. An allosteric network is thought to connect the proposed ratchet with a TVGG motif between helices 3 & 4 and gate residue M66 (Figure 5G), which together act as a hexamer/pentamer switch [46]. The TVGG motif alters interactions at capsomer interfaces and sits behind the three-fold interface where second-site mutant T216I is located. It has been proposed that IP6 binding regulates the TVGG switch, though the exact mechanism is unclear. Residue K25 is thought to play a particularly key role in IP6 binding within pentamers, because binding of a second IP6 molecule further down the pore allows closer packing within the pentamer. This may be the source of the propagated allosteric changes that favor a 3_{10} helix within the TVGG switch region [46]. Compensating mutations N21S and K25T could restore the closer

packing that is lost with binding of the second IP6 molecule, whilst mutation T216I may alter the allosteric network connecting the various interfaces. Importantly, the fact that second site mutants can rescue pentamer formation without restoring binding of the second IP6 molecule suggests that pentamer/hexamer thermodynamic equilibrium is not solely determined by IP6."

Minor comments:

Q5. Fig 2- legend "K25A but not R18 infectivity can be rescued." Do the authors mean R18 infectivity or R18 mutant (R18G) infectivity?

A5. We've simplified this title to: "K25A replication can be rescued by second-site mutations".

Q6. Page 6- "K25A but not R18A/S/T infectivity can be rescued by second site mutations." Considering the differential role of R18 and K25 in capsid assembly, it is not surprising that the second site mutations responsible for rescuing K25A could not restore R18 mutant infectivity. Maybe the authors can clarify if secondary mutations other than N21S/R18G or T216I/R18G can restore the effects of R18 mutations?

A6. We thank the reviewer for drawing our attention to this. Whilst we did not observe rescue of R18G with N21S or T216I, we cannot claim that no second-site compensatory mutations exist. We have therefore amended the title of this section and accompanying figure, as described above, to: "K25A replication can be rescued by second site mutations".

Reviewer 2

HIV-1 capsid is an assembly of 250 hexamers and 12 pentamers. Hexameric (also pentameric) capsomeres assemble a centralized pore that is bound by IP6 and has been postulated to act as a nucleotide importer by the James group. The formation of the central pore requires the coordination of 2 amino acids R18 and K25. Both amino acids coordinate the binding of capsid to IP6, which has been shown to stabilize the overall assembled capsid. The integrity of the capsid which is supported by IP6 binding is a key aspect that is necessary for post-entry roles of capsid, which includes serving the role of a container where protected viral DNA synthesis by reverse transcription could take place.

Here, Kleinpeter. et al present an elegant study that uses the forced evolution of viruses to compensate for the IP6 binding site mutant at the R18 and K25 amino acid residue of CA, the protein that makes up the HIV-1 conical capsid shell. Previously postulated importance of these residues is tested here elegantly, by forcibly evolving viruses under selective pressure in a replication assay. The authors conduct a series of structural, biochemical, and virological studies to illustrate the role of the second site suppressors on capsid assembly (in vitro, and in virions), capsid stability after membrane stripping and in cells. While R18 mutants failed to replicate, the K25A mutants evolved more efficiently to replicate in cultured cells by acquiring second-site suppressor mutants in the intra- and inter-capsomere interfaces. These compensatory substitutions in the K25 mutants overcome infectivity defect, while substitutions with T is more favorable than A at position K25. They find that R18A/T/S is a more rigid mutant, that fails to recover in replication assays. The reason why R18 mutants fails to replicate is likely because this mutation leads to predominantly pentameric assemblies, which is nicely demonstrated and supported by complementary structural studies conducted in this work. On the other hand, the acquisition of compensatory second-site suppressor mutations in K25A mutant viruses facilitate the assembly of conical capsids.

The main conclusion that R25 which was originally thought to be strictly essential can be by passed by second site suppressors on CA is well supported by the data and appropriately described in the main text. The broader implications includes 'conical morphology', and thus an optimal hexamer:pentamer

equilibrium is critical for the assembled HIV-1 capsid functions in mediating virus infection.

Notable questions that arose from my review include:

Q1. How does IP6 fit into this current story? Is IP6 even required for HIV-1 infection? Given that K25A/T216I among other mutants can effectively infect cells, do they still require IP6 binding and does dNTP import into the core of the virus still go through the central pore formed in K25A/T216I mutant hexamer? Do the authors perceive that dNTP import into the core can also occur through other mechanisms? It will be nice to see endogenous reverse transcription (ERT) in the identified new capsids.

A1. Previously published data suggests that IP6 is required for HIV-1 infectivity, because it is needed for correct capsid assembly (Dick et al Nature 2018, Renner et al NSMB 2023, Mallery et al Sci Adv 2021). IP6 also greatly increases capsid stability (Mallery et al eLife 2018), although only selected capsid mutants have been shown to be sensitive to IP6 depletion in target cells (Sowd et al PLOS Path 2023). Whether this is because residual cellular IP6 can still be bound by wild-type capsids or because IP6 is not required post-fusion is unknown. Our data show that K25A/T216I CA still binds nucleotides and IP6 (Figure 4A-C) and that IP6 is still needed for in vitro capsid assembly (Figure 5C&D). Assembled T216I and K25A/T216I capsids are also stabilized by IP6 in a similar manner to wild-type (Figure 6E). While we did not look at endogenous reverse transcription (ERT), we did compare K25A/T216I reverse transcription to WT in cells during infection and observed similar DNA synthesis kinetics from early through to late transcripts (Figure 4D). These results suggest that K25A/T216I is no different from WT in their dependence on IP6 and dNTP import through the central pore. However, just as with WT, it isn't currently possible to directly demonstrate these dependencies in infected cells.

We thank the reviewer for raising these points and have taken the opportunity to add the following sentences into our revised manuscript to address them:

“However, the fact that the rescued K25A mutants still bind nucleotides and IP6 (Figure 4A-C), use IP6 for in vitro capsid assembly (Figure 5C&D), are stabilized by IP6 similar to wild-type (Figure 6E) and have similar reverse transcription kinetics (Figure 4D) suggest that the second-site compensatory mutations do not rescue K25A replication by conferring IP6 independence. Analysing the location of compensating mutations also does not suggest that they generate a new pore for nucleotide import.”

Q2. R18G forms pentameric capsids, and K25A forms hexameric capsids – so R18 is required for hexamer formation and K25 for pentamer formation. The failure of R18 mutants to replicate highlights importance of hexamers in capsid assembly. However, how does secondary suppressor mutations in K25A/T mutants restore pentamer formation? The answer to this question, while is somewhat addressed by using models to show proximity of second-site suppressors to the K25 amino acid, this is at best hypothetical.

A2. Our data suggest that second site mutations restore pentamer formation because of their ability to restore conical capsid assembly in vitro. Mutations like N21S and T216I must allow pentamer formation because, as the reviewer correctly points out, K25A forms only hexameric tubes in vitro. From a purely thermodynamic perspective this can only be because second site mutations make pentamer formation more favourable. This is likely achieved by N21S and T216I altering packing at their respective interfaces. We have used models to show what residues are located nearby N21S and T216I but we accept we have no data directly implicating specific interactions or quantifying their relative contributions to favouring pentamer formation. This would be nice to know but difficult to achieve as we could never be sure that any interface mutations are only affecting pentamers and not hexamers or other properties associated with gag or capsid.

We have added the following into the discussion to address these important points:

“Exactly how second site mutations promote pentamer formation is unclear but is likely through alterations at their respective interfaces that make pentamers more energetically favourable. In the case of N21S this may be through optimising contacts between monomers in and around the central pore. Closer packing is required in the pentamer between helices 1, 2 and 3 and the exchange of asparagine for serine may favour these interactions. N21S is also located next to the proposed ratchet that is thought to propagate changes allosterically to the ‘TVGG’ motif that switches between conformations favoring pentamer or hexamer NTD-NTD and NTD-CTD packing[46]. For T216I, this is likely through changes at the 2- and 3-fold symmetry axis that promote the closer packing of monomers as required within pentamers.”

We have also expanded the Results section and added a new figure (5G) to show that second-site mutations are located next to the TVGG motif and ratchet mechanism.

Q3. Additional question that arose: Does the central hexamer/pentamer ring coordinating IP6 still formed following K25A/T + T216I substitutions? And is IP6 still required to stabilize these capsids? remains unclear. I say this because in Fig. 6E, while IP6’s role in stabilization of WT, N21S, N21S/K25A is somewhat shown (although distribution seems weird, in all of these the fold-difference of IP6 stabilization appears similar), there seem to be less of an effect of IP6 on T216I, and K25A/T216I. I would encourage the authors to consider re-plotting this graph as % of intact cores at 30 minutes by normalizing to initial intact cores seen immediately after SLO treatment (% of initial timepoint), and plotting bar graphs of fold-differences of IP6 stimulated stability changes may provide more insights and better interpretations. Also, I am not sure the TRIM5 abrogation assay reports stability of capsids in cells, but rather TRIM binding capacity.

A3. Permeabilization of virions by SLO is not immediate, which is why we use a 30 minute timepoint and normalize to untreated virions rather than a 0 minute timepoint. At 30 minutes we compare \pm SLO to determine the proportion (as a %) of capsids that are still intact. The no IP6 gives us a measure of intrinsic stability whilst the + IP6 measures stability in the presence of IP6. We apologize for any confusion and have clarified the figure legend for Figure 6E, as this was unclear. We agree with the reviewers point that there are differences in the IP6 effect between mutants. This is likely because there are differences in their intrinsic stability, as shown in the no IP6 condition (eg N21S is slightly less stable than WT, and T216I more stable, in the absence of IP6). Nevertheless, the TIRF data support the central conclusion that IP6 increases the stability of all mutants except K25A alone. Other data also support that the IP6 binding site provided by the central pore is maintained in K25A/T216I as the mutant still binds IP6 (Figure 4A&B) and IP6 is still needed to drive conical capsid assembly in vitro (Figure 5C&D). TRIM5 has been widely used as a proxy for capsid stability because it is a capsid sensor that no longer binds when the capsid uncoats or collapses (eg PMID 29187540 & 16624363) but we do appreciate it is an indirect assay.

Q4. Regarding completeness of the implications of new substitutions: While infectivity measurements are provided to show that second-site suppressors can overcome K25A mutation defects, what happens to the second-site suppressor containing K25A capsids in cells remain unclear. The cell biology aspect of these new capsid variants is missing. The kinetics of viral DNA synthesis is shown seems like no apparent change, how about nuclear import, and integration? Are there effects in primary cells in terms of sensing where host-factor usage is critical? It is known that R18A cannot bind a host-factor PQBP1, does K25A/T216I or in combination with N21S impose defects in this interactions?

A4. We agree it is important to consider how efficiently second-site suppressors restore HIV-1 K25A fitness in a physiologically meaningful cellular context. We note that the second-site suppressors arose during studies of replication-competent virus in three different T cell lines. These experiments (Figure 2) show that second-site mutations are sufficient to restore K25A replication to close to wild-type replication. A significant defect in any of the individual steps (eg integration) would be expected to delay or prevent replication.

Minor:

Pg.3 Line 43 "200 hexamers" is this a correct number? Perhaps provide a citation. Published literature suggests 240 or 250 hexamers.

We have provided a citation and corrected to 250 hexamers.

Pg. 3 Line 48: "capsids remain intact" I would suggest using the phrase "remain largely intact" here, since this is still an unresolved issue.

We have removed this phrase and now just say it plays a key role.

Pg. 6 Line 120-123: "The fact that K25A has an under-assembly defect, while R18G suffers from over-assembly is consistent with the two residues playing different roles in capsid assembly and morphology" – is it possible these mutants affect virus budding and release? Please include data for virus production and associated western blots

We have included western blots from three independent viral production experiments and calculated the virus release efficiency. This data has been included as new Supplementary Figure 2.

Pg. 8 Line 185: "fluorescent ATP analog" please describe here how this incorporates into the thermal stability measurements. Why fluorescent? Can ATP alone be used?

We apologise for the confusion - fluorescent ATP is used in fluorescent binding experiments and not in thermal stability measurements. To clarify this we have added the phrase "in a fluorescence polarization binding assay" into the main text and the phrase "as measured by fluorescence polarization binding assays" into the figure legend.

Pg. 8 Line 201-203: "These data suggest that K25 is not required for dNTP import through the pore, although the inability of second-site mutations N21S or T216I to fully restore the accumulation of viral DNA to WT levels suggests that either import is slightly less efficient, or capsids are slightly less stable." Check for sense. To this reviewer the suppressor mutants seems to recover defects in vDNA synthesis – fully and is correlated to infectivity data. Not sure what is being highlighted here.

We have amended this sentence to: "These data suggest that K25 is not required for dNTP import through the pore."

Line 315: T216I was also isolated from transmitter founder viruses and in patient samples (Siddiqui et al.,), is it not interesting that this mutant also shows up in the forced evolution?

We thank the reviewer for making this point and have added the fact that T216I is found in transmitter founder viruses to our discussion.

Fig. 6C-E: Core stability assay using iGFP: "Capsid particles from three to five images were acquired for each condition and normalised to untreated control" please elaborate. (1) was the particle distribution

similar when imaging intact virions, (2) how many iGFP containing cores were retained following SLO treatment, and (3) what was the % that remained stable from this pool.

We apologise that our description of this assay was confusing and poorly written. We have corrected the figure legend as follows: “The number of capsid particles from three to five 88 μm^2 images (typically 500-1000 particles/image) were counted for each condition. Graph shows the fraction of intact capsids in +SLO conditions (as a percentage of the mean under -SLO conditions) from multiple images from three independent experiments.” The particle distribution and means for different mutants were different, indicative of differences in intrinsic stability as expected. The percent of iGFP-containing cores following SLO treatment was approximately 30% in the absence of IP6 and 85% in the presence of IP6 for wild-type virus. For all mutants except K25A there was a statistically significant increase in the number of cores in the presence vs absence of IP6.

From Fig. 6C-E legends: “Graph shows particle numbers from multiple images from three independent experiments. Multiple unpaired T tests were used to compare conditions to IP6.”- As of now, I see a distribution plot of particle numbers, but what seems like a distribution plot of fluorescence. If the numbers were used from independent experiments one would expect to see a few dots and error bars.

We have re-drawn the graph in Figure 6E to show each data point plus the mean. To be clear, each data point represents the results from a single 88 μm^2 image. Each experiment included at least three images and the experiment was repeated at least three times.

The in vitro measurements of stability seem to suggest that K25A is just as stable as the WT capsid (Fig. 6E). Yet it fails to overwhelm TRIM restriction in cells (Fig. 6F). Does K25A fail to abrogate TRIM activity because it does not bind the restriction factor? I am unsure if this is the right assay to demonstrate capsid stability in cells. Also, a control using unstable capsids seems to be missing.

When comparing the in vitro TIRF data with the cellular TRIM5 data, the + IP6 condition should be used. This is because IP6 is present in cells. In the presence of IP6 there is a stability difference between K25A and WT. K25A fails to abrogate TRIM5 activity in cells because its capsid is unstable.

REVIEWERS' COMMENTS

Reviewer #1 (Remarks to the Author):

Minor Comments for revision that can be added to improve discussion:

1. It would be helpful to the readers for the authors to include a few sentences in the discussion on a likely mechanistic model for the following statement: “N21S and T216I are capable of largely rescuing the defect in infection 175 caused by K25A in isolation, whereas others, such as A105T, G208R and G208R, G225S only show a substantial rescue when combined with K25T (Figure 3B).” (Page 8, line 175). Which interfaces in CACTD are affecting the hexamer/pentamer equilibrium? Is there a long-range allostery in all these mutants?
2. Related to the question above: how do these second site mutations modulate the structural state and order of the IP6-binding pore itself? For example, what are the (possible) residue-residue interactions these second-site mutations modulate?
3. Page 9, line 225: “Assembly occurred with similar kinetics for all mutants, except K25T and K25T/T216I, which assembled more rapidly than WT (Figure 5A).” – The kinetics of assembly is interesting to many readers. It would be good to include a short discussion on the structure-kinetics relationship in different mutants.
4. While the authors show that second-site suppressors can overcome K25A mutation defects, how may these second-site mutations impact host factor interactions that are key for the nuclear import of intact capsids?

Reviewer #2 (Remarks to the Author):

My concerns have been appropriately addressed.

Reviewer #1:

Q1. It would be helpful to the readers for the authors to include a few sentences in the discussion on a likely mechanistic model for the following statement: "N21S and T216I are capable of largely rescuing the defect in infection 175 caused by K25A in isolation, whereas others, such as A105T, G208R and G208R, G225S only show a substantial rescue when combined with K25T (Figure 3B)." (Page 8, line 175). Which interfaces in CACTD are affecting the hexamer/pentamer equilibrium? Is there a long-range allostery in all these mutants?

A1. We have provided an additional paragraph in the discussion to address these suggestions.

Q2. Related to the question above: how do these second site mutations modulate the structural state and order of the IP6-binding pore itself? For example, what are the (possible) residue-residue interactions these second-site mutations modulate?

A2. Mapping the relative contribution of each mutant to hexamer/pentamer equilibrium will require further structural work. We have however provided speculation in the manuscript and in the discussion paragraph (see above).

Q3. Page 9, line 225: "Assembly occurred with similar kinetics for all mutants, except K25T and K25T/T216I, which assembled more rapidly than WT (Figure 5A)." – The kinetics of assembly is interesting to many readers. It would be good to include a short discussion on the structure-kinetics relationship in different mutants.

A3. While a detailed structure-kinetics analysis would be interesting, we would prefer to wait for detailed structural information for each specific mutant, which will require a further study.

Q4. While the authors show that second-site suppressors can overcome K25A mutation defects, how may these second-site mutations impact host factor interactions that are key for the nuclear import of intact capsids?

A4. We have included a discussion and references in a new paragraph to address possible impacts on host factor interactions.